# ARID1A loss in adult hepatocytes activates β-catenin-mediated erythropoietin transcription

Rozenn Riou[1,2,3], Meriem Ladli[3], Sabine Gerbal-Chaloin[4], Pascale Bossard[2,3], Angélique Gougelet[1,2,3], Cécile Godard[1,2,3], Robin Loesch[1,2,3], Isabelle Lagoutte[3,5], Franck Lager[3,5], Julien Calderaro[6,7], Alexandre Dos Santos[8], Zhong Wang[9], Frédérique Verdier[3], Sabine Colnot[1,2,3]*

[1]INSERM, Sorbonne Université, Université de Paris, Centre de Recherche des Cordeliers (CRC), Paris, France; [2]Equipe labellisée Ligue Nationale Contre le Cancer, Paris, France; [3]INSERM, CNRS, Institut COCHIN, Paris, France; [4]INSERM U1183, Université Montpellier, Institute for Regenerative Medicine & Biotherapy (IRMB), Montpellier, France; [5]Plateforme d'Imageries du Vivant de l'Université de Paris, Paris, France; [6]INSERM, Université Paris-Est UPEC, Créteil, France; [7]Department of Pathology, Henri Mondor Hospital, Créteil, France; [8]INSERM, Paul-Brousse University Hospital, Hepatobiliary Centre, Villejuif, France; [9]Department of Cardiac Surgery Cardiovascular Research Center, University of Michigan, Ann Arbor, United States

**Abstract** Erythropoietin (EPO) is a key regulator of erythropoiesis. The embryonic liver is the main site of erythropoietin synthesis, after which the kidney takes over. The adult liver retains the ability to express EPO, and we discovered here new players of this transcription, distinct from the classical hypoxia-inducible factor pathway. In mice, genetically invalidated in hepatocytes for the chromatin remodeler *Arid1a*, and for *Apc*, the major silencer of Wnt pathway, chromatin was more accessible and histone marks turned into active ones at the *Epo* downstream enhancer. Activating β-catenin signaling increased binding of Tcf4/β-catenin complex and upregulated its enhancer function. The loss of *Arid1a* together with β-catenin signaling, resulted in cell-autonomous *EPO* transcription in mouse and human hepatocytes. In mice with *Apc-Arid1a* gene invalidations in single hepatocytes, Epo de novo synthesis led to its secretion, to splenic erythropoiesis and to dramatic erythrocytosis. Thus, we identified new hepatic *EPO* regulation mechanism stimulating erythropoiesis.

*For correspondence: sabine.colnot@inserm.fr

Competing interests: The authors declare that no competing interests exist.

## Introduction

Chromatin dynamics strongly modulates gene expression, and the liver is a prominent tissue in which chromatin opening is a pre-pattern for cell fate programming (*Zaret, 2016*). ARID1A, 'AT-rich interacting domain containing protein 1A', is a BAF (BRG1-associated factors) subunit of the highly evolutionarily conserved SWI/SNF chromatin remodeling complexes. These complexes use the energy of ATP hydroxylation to reposition, eject, or exchange nucleosomes and thus modulate DNA accessibility (*de la Serna et al., 2006*). They are essential for the regulation of gene expression and are involved in several cellular functions, such as differentiation, development, proliferation, DNA repair, and adaptation to the extracellular environment (*Kadoch et al., 2016*). Recently, mutations in chromatin modifying factors have been identified in several types of cancer (*Kadoch et al., 2016*).

In the adult mouse liver, Arid1a has been shown to play a role in liver regeneration and in tumorigenesis (*Sun et al., 2018*; *Sun et al., 2016*). In human hepatocellular carcinoma (HCC), the most common primary liver cancer (*Torre et al., 2016*), ARID1A is the chromatin modifier gene the most frequently inactivated (>13% of HCCs). These mutations are preferentially found in HCC with activating mutations of the *CTNNB1* gene encoding β-catenin, accounting for one third of HCC (*Guichard et al., 2012*; *Rebouissou et al., 2016*). This suggested a potential link between Wnt/β-catenin pathway and ARID1A for the regulation of hepato-specific gene expression programs involved in liver pathophysiology.

In the adult liver, the Wnt/β-catenin pathway can induce both physiological and oncogenic effects (*Cavard et al., 2008*; *Colnot, 2016*; *Monga, 2015*). Such signaling is restricted to the hepatocytes surrounding the central vein, the so-called pericentral hepatocytes, where it is activated by nearby endothelial Wnt and R-Spondin ligands (*Planas-Paz et al., 2016*; *Benhamouche et al., 2006*). β-catenin transcriptionnally patterns the liver to ensure its pericentral metabolic functions (*Gougelet et al., 2014*; *Torre et al., 2011*). A genetically engineered panlobular activation of the Wnt/β-catenin pathway quickly induced a pericentral-like liver phenotype and hepatomegaly, resulting in mouse death (*Benhamouche et al., 2006*). Additionally, the focal activation of β-catenin in vivo in single murine hepatocytes is oncogenic, leading to the development of β-catenin-activated liver tumors (*Colnot et al., 2004*). We used transcriptomic and metabolomic approaches and showed that the genetic program expressed in β-catenin-activated liver is similar to the oncogenic signature found in human HCC harboring activating β-catenin mutations (*Gougelet et al., 2014*; *Gougelet et al., 2019*; *Senni et al., 2019*).

When activated, β-catenin translocates into the nucleus and interacts with its co-factor Tcf4 to bind Wnt-responsive elements (WRE) located in the vicinity of target genes (*Gougelet et al., 2014*). Chromatin remodeling processes have been shown to unlock chromatin over WREs, allowing β-catenin to dictate specific transcriptomic programs (*Mosimann et al., 2009*). Given the frequent inactivation of *ARID1A* in *CTNNB1*-mutated liver tumors, our aim was to determine in mice whether and how the loss of the chromatin remodeler Arid1a cooperates with β-catenin to impact on mouse liver pathophysiology. We used transgenic murine models in which the main brake of the Wnt/β-catenin pathway, the tumor suppressor *Adenomatous polyposis coli* (*Apc)* (*Colnot et al., 2004*) and/or *Arid1a* (*Gao et al., 2008*) are lost in adult hepatocytes. We unexpectedly revealed a novel major function of ARID1A and the Wnt/β-catenin pathway in regulating *EPO* expression and adult erythropoiesis.

## Results

### Emergence of peliosis-like regions in the liver of [*Apc-Arid1a*]$^{\text{ko-focal}}$ mice

We investigated the effects of the loss of the chromatin remodeler Arid1a in a context of focal and aberrant β-catenin activation. To do so, we injected transgenic mice carrying *Apc* and/or *Arid1a* floxed genes with a low dose of Cre-expressing Adenovirus (AdCre) known to mainly target the liver (*Colnot et al., 2004*). In *Apc*-floxed mice, we previously showed that this dose was sufficient to induce β-catenin activation in single hepatocytes and promote tumorigenesis without killing the mice (*Colnot et al., 2004*). Accordingly, this injection in compound *Apc/Arid1a*-floxed mice inactivated both *Apc* and *Arid1a* genes in approximately 20% of hepatocytes ([*Apc-Arid1a*]$^{\text{ko-focal}}$ mice, *Figure 1a*, *Figure 1—figure supplements 1*).

Surprisingly, an ultrasound follow-up showed the development of striking echogenic features in [*Apc-Arid1a*]$^{\text{ko-focal}}$ mouse livers from 5 months after AdCre injection (*Figure 1c*, *Figure 1—figure supplements 2a*). We revealed after dissection that these livers harbored numerous and irregular dark red to black vascular lesions (*Figure 1b*). After 10 months, all [*Apc-Arid1a*]$^{\text{ko-focal}}$ mice (n = 24) exhibited blood-filled lacunar spaces (*Figure 1c*), as well as hepatomegaly (*Figure 1—figure supplements 1a*). We did not however observe such phenotypic abnormalities in the [*Apc*]$^{\text{ko-focal}}$ (n = 13), 18 [*Arid1a*]$^{\text{ko-focal}}$ (n = 18), or control (n = 10) mice studied. [*Apc-Arid1a*]$^{\text{ko-focal}}$ mice exhibited 50% and 100% mortality at 10 and 14 months, respectively (*Figure 1d*). In dying mice, we discovered that the whole liver was diseased and dark red in color. Indeed, the liver was filled with blood, harboring large necrotic areas with no remaining healthy zones (*Figure 1d*, inset).

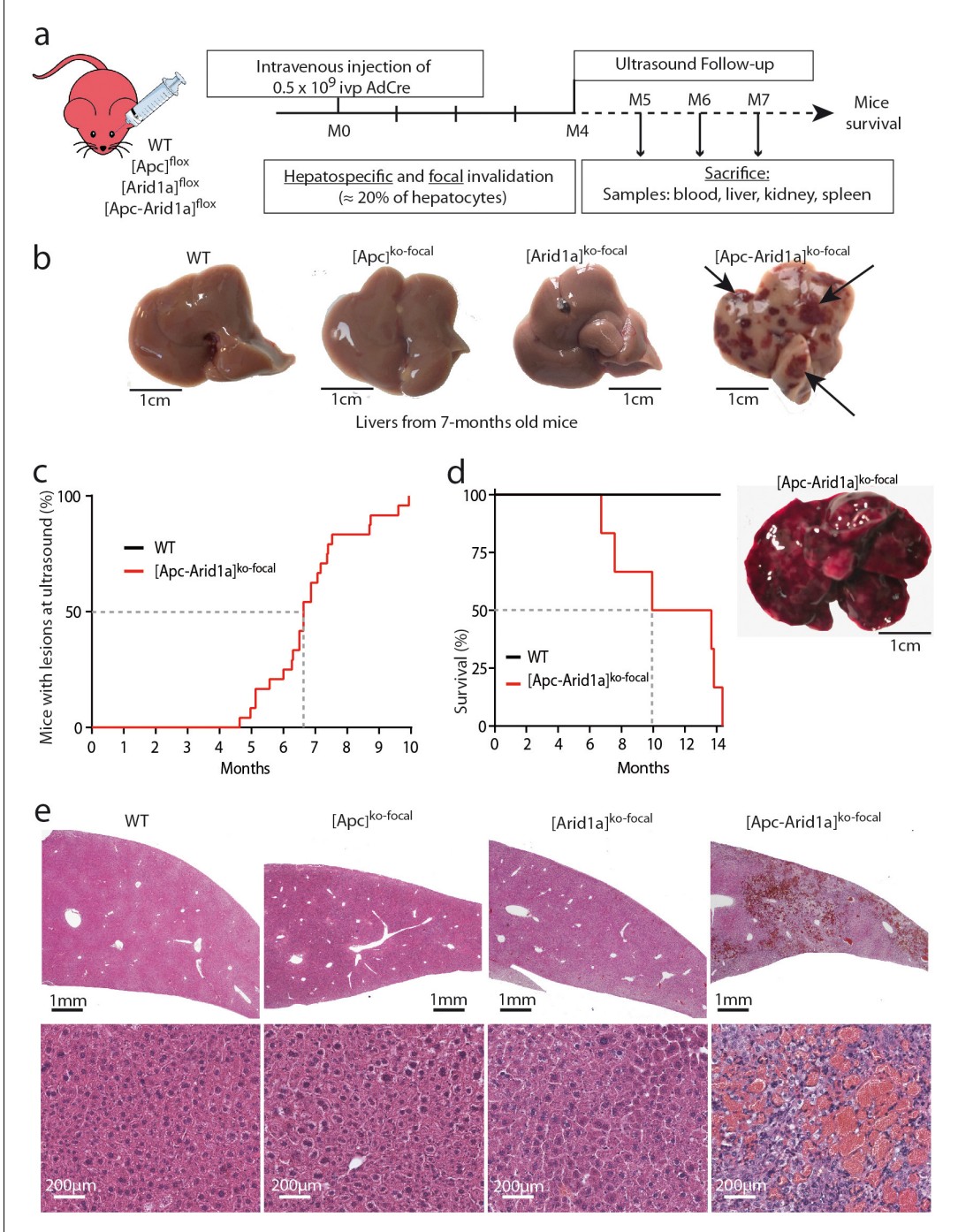

**Figure 1.** Development of peliosis-like regions after hepato-specific and focal *Arid1a* and *Apc* inactivation. (a) Cre-loxP-generated hepatocyte-specific and inducible inactivation of *Apc* and/or *Arid1a* in 20% of hepatocytes after retro-orbital injection of infectious viral particles (ivp) of adenovirus encoding Cre recombinase (AdCre). The resulting mice are referred to as [*Apc-Arid1a*]^ko-focal, [*Apc*]^ko-focal, and [*Arid1a*]^ko-focal. (b) Gross examination of mouse livers, 7 months after AdCre injection. Livers from [*Apc-Arid1a*]^ko-focal mice had an irregular shape and a rough surface, with multiple dark red zones (indicated by arrows). (c) Incidence of hepatic lesions detected in WT (n = 10) and [*Apc-Arid1a*]^ko-focal (n = 24) mice by ultrasonography. (d) Kaplan-Meier estimated survival curves of WT and [*Apc-Arid1a*]^ko-focal mice over 15 months. n = 6 for each group. Inset: Liver of one mouse at necropsy (13 months after AdCre injection, representative of the three analyzed mice). (e) Hematoxylin Eosin (HE)-stained sections of mouse livers at 7 months post-injection. Large vascular spaces filled with blood cells were observed only in [*Apc-Arid1a*]^ko-focal livers. Related data are found in *Figure 1—figure supplements 1–4*, and source data in '*Figure 1—source data 1*; *Figure 1—figure supplement 1—source data 1*; *Figure 1—figure supplement 3—source data 1*'.

The online version of this article includes the following source data and figure supplement(s) for figure 1:

*Figure 1 continued on next page*

*Figure 1 continued*

**Source data 1.** Emergence of peliosis (*Figure 1c*) and survival curve (*Figure 1d*).

**Figure supplement 1.** Focal inactivation of *Apc* and/or *Arid1a* genes in mouse liver.

**Figure supplement 1—source data 1.** Liver to body weight ratios (*Figure 1—figure supplements 1a*) and expression of Glul and Axin2 mRNAs (*Figure 1—figure supplements 1b*).

**Figure supplement 2.** Ultrasound features of livers from seven-month-old [*Apc-Arid1a*]$^{ko-focal}$ mice.

**Figure supplement 3.** Blood vessel enrichment and angiogenesis in [*Apc-Arid1a*]$^{ko-focal}$ livers.

**Figure supplement 3—source data 1.** qPCR expression of angiogenic mRNAs (*Figure 1—figure supplements 3c*).

**Figure supplement 4.** Hepatocarcinogenesis in β-catenin-activated and Arid1a-null context.

Histologically, the diseased [*Apc-Arid1a*]$^{ko-focal}$ liver showed abnormal blood vessels that were partially or completely full of red blood cells (RBCs) (*Figure 1e*, *Figure 1—figure supplements 3a*), associated with sinusoidal dilatation and liver cell dropout. Additionally, using microbubble-assisted ultrasound, we showed a decrease in hepatic vascular perfusion within echogenic areas, illustrating hence a vascular liver disease (*Figure 1—figure supplements 2a*). We thus characterized these areas with dramatic histological features as peliosis-like areas, similar to the human vascular disease, peliosis.

In accordance with previous results (*Colnot et al., 2004*), β-catenin-activated liver tumors developed in 92% of [*Apc*]$^{ko-focal}$ mice (*Figure 1—figure supplements 4a*). Here, only 8% of [*Apc-Arid1a*]$^{ko-focal}$ mice developed liver tumors which were both β-catenin-activated and *Arid1a*-invalidated (*Figure 1—figure supplements 4a-c*), suggesting that Arid1a loss suppresses the tumorigenic effect of activated Wnt/β-catenin signaling in the liver. However, this model was not appropriate for assessing the effects of *Arid1a* loss on Wnt/β-catenin-dependent hepatocarcinogenesis in these mice, given the emergence of peliosis and lethality at a stage preceding or overlapping the expected tumor initiation phase (*Figure 1c*, *Figure 1—figure supplements 4a-c*).

We reveal here that β-catenin activation and Arid1a loss cooperate to induce a dramatic hepatic peliosis and lethality in the mouse.

## Hepatic loss of both *Arid1a* and *Apc* results in erythrocytosis linked to de novo transcription of *Epo*

We performed transcriptomic microarray analysis of micro-dissected [*Apc-Arid1a*]$^{ko-focal}$ livers (*Figure 2a*). Firstly, gene set enrichment analysis (GSEA) revealed transcriptional signatures linked to angiogenesis and the Erythropoietin (EPO) pathway in peliosis-like areas relative to adjacent regions (*Figure 2b,c* and *Figure 2—figure supplements 1*). Additionally, these peliosis-like regions showed a Wnt/β-catenin transcriptional signature, revealing enrichment of β-catenin-activated cells within these areas.

We then analyzed the hematological parameters and complete blood cell counts from peripheral blood. RBC counts, as well as hematocrit and hemoglobin levels, were significantly higher in [*Apc-Arid1a*]$^{ko-focal}$ mice than in control or single knockout mice (*Figure 2d*). This confirmed that blood erythrocytosis corresponded to erythrocyte overload.

The production of RBCs, known as erythropoiesis, is a dynamic process requiring the orchestration of specific molecular mechanisms (*Nogueira-Pedro et al., 2016*). These include for example the key EPO cytokine, a circulating glycoprotein hormone (*Jelkmann, 2007*). In mouse embryos, hepatoblasts are the primary source of Epo. In adults, the site of production switches from the liver to the kidney (*Weidemann and Johnson, 2009*), but the adult liver can still produce Epo (*Suzuki, 2015*). To determine whether erythrocytosis in [*Apc-Arid1a*]$^{ko-focal}$ mice could be due to dysregulation of this key hematological regulator, we examined *Epo* transcript and protein levels within the entire liver and the plasma fraction, respectively. We detected a marked reactivation of *Epo* expression in [*Apc-Arid1a*]$^{ko-focal}$ livers, whereas no Epo expression was detected in either single knockout or control livers (*Figure 2e*). This was associated with distinctly higher Epo protein levels in the plasma of [*Apc-Arid1a*]$^{ko-focal}$ mice (*Figure 2f*). We confirmed that plasma Epo derived from the liver as we observed no change in Epo transcription in the kidneys of [*Apc-Arid1a*]$^{ko-focal}$ mice (*Figure 2e*). Interestingly, we saw no changes in Epo mRNA levels in human HCC harboring the compound *CTNNB1/ARID1A* mutations (*Figure 1—figure supplements 4d*).

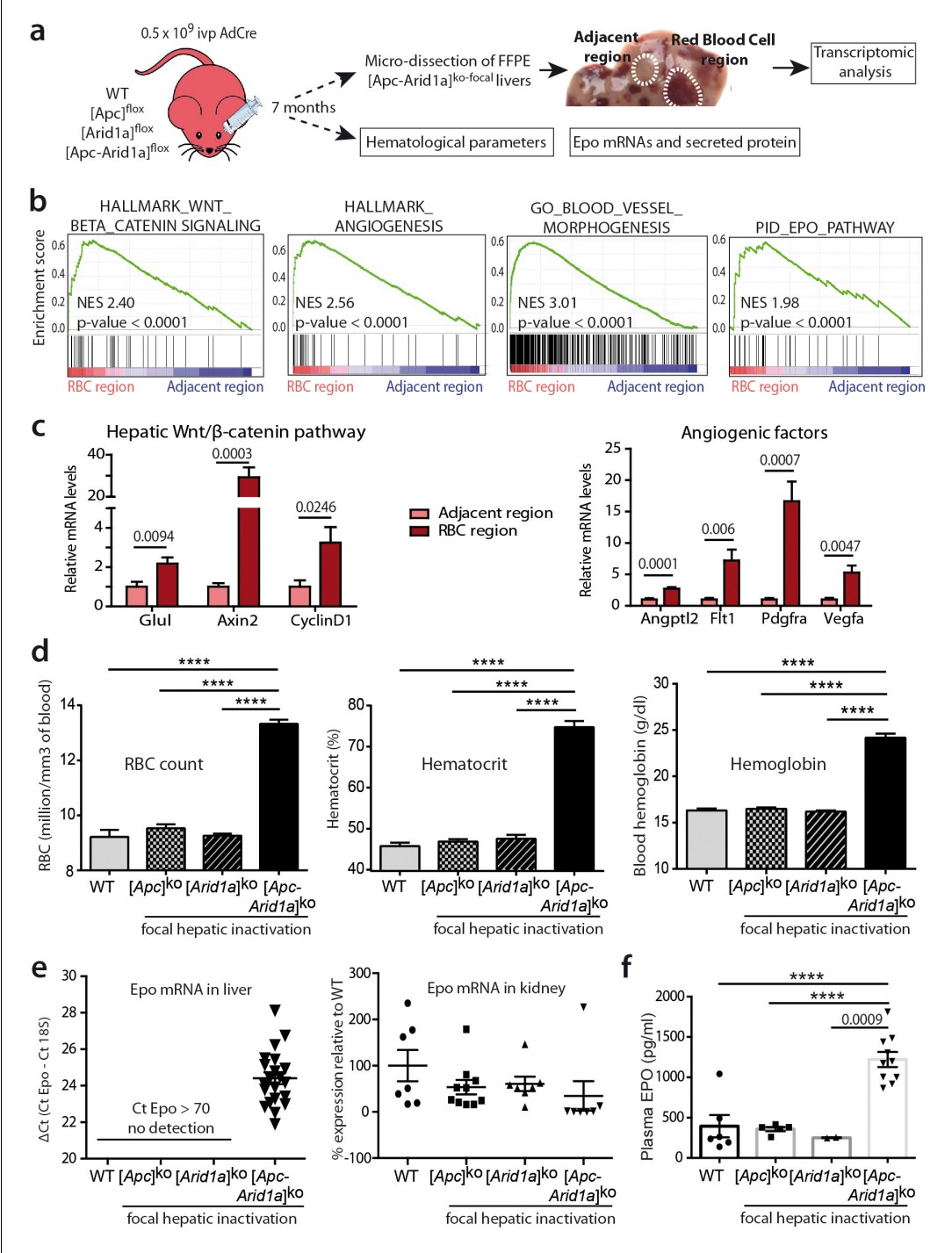

**Figure 2.** Hepatic peliosis has 'angiogenic' and 'erythropoietin' transcriptional signatures, linked to a systemic erythrocytosis and to de novo hepatic *Epo* expression in [*Apc-Arid1a*]ko-focal mice. (**a**) Experimental strategy; (**b**) Transcriptomic gene-set enrichment analysis (GSEA) of hepatic peliosis (n = 4) relative to adjacent regions (n = 4) of [*Apc-Arid1a*]ko-focal mice. (**c**) Quantitative RT-PCR showing relative expression of mRNAs for positive targets of hepatic Wnt/β-catenin pathway and angiogenic factors in hepatic peliosis (n = 10) compared to adjacent regions (n = 10) of [*Apc-Arid1a*]ko-focal mice (unpaired t test analysis); (**d**) Hematological parameters from WT (n = 7), [*Apc*]ko-focal (n = 12), [*Arid1a*]ko-focal (n = 19), and [*Apc-Arid1a*]ko-focal (n = 20) mice (One-way ANOVA analysis). (**e**) Evaluation of erythropoietin (*Epo*) mRNAs by quantitative RT-PCR in the livers analyzed by the ΔCt technique and expressed relative to those for 18S RNA for the liver, and as relative levels in the kidney (One-way ANOVA analysis). (**f**) Plasma EPO concentrations at sacrifice (WT (n = 6), [*Apc*]ko-focal (n = 5), [*Arid1a*]ko-focal (n = 2), and [*Apc-Arid1a*]ko-focal (n = 10)). Exact p-values are mentioned, ****p<0.0001. Related data are found in *Figure 2—figure supplements 1* and source data in '*Figure 2—source data 1*'.

*Figure 2 continued on next page*

Figure 2 continued

The online version of this article includes the following source data and figure supplement(s) for figure 2:

**Source data 1.** Gene expression (*Figure 2c, e*) and hematological parameters (*Figure 2d*).

**Figure supplement 1.** Peliosis-like regions from [*Apc-Arid1a*]$^{ko-focal}$ livers are enriched for 'Endothelium development' and 'Erythrocyte homeostasis' transcriptional signatures.

---

Overall, our findings demonstrate that simultaneous *Arid1a* loss and β-catenin activation in single hepatocytes, occurring in a physiological but non-cancerous context, are responsible for a major hematological disorder that is linked to de novo expression and subsequent secretion of hepatic Epo.

## Erythropoiesis is induced in the spleens of [*Apc-Arid1a*] $^{ko-focal}$ mice

To determine the site of pathological production of the RBCs observed in [*Apc-Arid1a*]$^{ko-focal}$ mice, we examined the liver, bone marrow (BM), and spleen; these are the three major organs responsible for erythropoiesis during embryogenesis (*Suzuki et al., 2011*), adult life (*Suzuki, 2015*), and stress responses in mice (*Perry et al., 2009*), respectively. Firstly, gross dissection of [*Apc-Arid1a*]$^{ko-focal}$ mice revealed a marked splenomegaly (*Figure 3a,b*). Histological sections from [*Apc-Arid1a*]$^{ko-focal}$ spleens showed prominent expansion of the red pulp with a predominance of erythroblasts relative to control spleens (*Figure 3c*).

We additionally quantified erythroid precursors in the liver, BM, and spleen by flow cytometry (corresponding to the TER119$^+$/CD71$^+$ cell population). In [*Apc-Arid1a*]$^{ko-focal}$ liver non-parenchymal cells (NPCs) relative to controls, there was no difference in TER119$^+$/CD71$^+$ progenitors revealing no intra-hepatic erythropoiesis (*Figure 3d,e*). However, there was a striking increase in the RBC population (TER119$^+$/CD71$^-$). This liver erythrocytosis was confirmed by immunostaining of the hemoglobin subunit beta (HBB) in liver tissue sections, showing that RBCs, but not erythroblasts, accumulated in these livers (*Figure 3—figure supplements 1*). In addition, TER119$^+$/CD71$^+$ cell populations were similar in the BM of [*Apc-Arid1a*]$^{ko-focal}$ and control mice, whereas we found threefold more erythroid precursors in [*Apc-Arid1a*]$^{ko-focal}$ spleens than in control spleens (*Figure 3d,e*). This suggested that RBC overproduction came from splenic and not from medullary or hepatic erythroblasts. We then analyzed the ability of erythroid progenitors to expand by in vitro quantification of erythroid colony-forming units (CFU-E) from spleen cells, BM cells, and liver NPCs. We confirmed the presence of erythroid progenitors in the BM and spleens of control mice after 3 days of culture in the presence of EPO, and their absence in control liver NPCs (*Figure 3f*). After EPO treatment, the spleens of [*Apc-Arid1a*]$^{ko-focal}$ mice contained 13-fold more CFU-E than control spleens (*Figure 3f*). This was not the case for the liver or BM. Finally, there were higher mRNA levels of erythropoiesis-related signaling components (*Nogueira-Pedro et al., 2016*) in the spleens of [*Apc-Arid1a*]$^{ko-focal}$ mice than those of control or single knockout mice (*Figure 3g*), including that of the Epo receptor.

Overall, these data show a strong increase in erythropoiesis and erythrocyte progenitors in the spleens of [*Apc-Arid1a*]$^{ko-focal}$ mice.

## Blocking Epo signaling reverses erythrocytosis and splenic erythropoiesis, but maintains liver angiogenesis

We analyzed the role of Epo in the dramatic phenotype of [*Apc-Arid1a*]$^{ko-focal}$ mice. We used an anti-EPO blocking serum which neutralizes soluble erythropoietin in mice (*Mastrogiannaki et al., 2012*). Anti-Epo treatment restored the hematocrit level of [*Apc-Arid1a*]$^{ko-focal}$ mice to that of untreated controls (*Figure 4a*, *Figure 2d*), showing a reversal of blood erythrocytosis. We quantified 10-fold less erythroid precursors and a lower mRNA expression of erythropoiesis factors in the spleen of anti-Epo treated [*Apc-Arid1a*]$^{ko-focal}$ mice compared to untreated mice (*Figure 4b–d*).

EPO is a pleiotropic growth factor which can stimulate vessel growth through an autocrine and/or paracrine loop (*Kimáková et al., 2017*). We tested the attractive possibility that hepatocyte-secreted Epo in [*Apc-Arid1a*]$^{ko-focal}$ mouse livers regulates RBC homing to the liver through increased angiogenesis. Liver tissue sections showed that blood vessels contained less RBCs in anti-EPO treated [*Apc-Arid1a*]$^{ko-focal}$ mice compared to untreated mice (*Figure 4e*, *Figure 4—figure supplements 1*), and these livers harbored less TER119$^+$/CD71$^-$ mature RBCs (*Figure 4f–g*). Despite

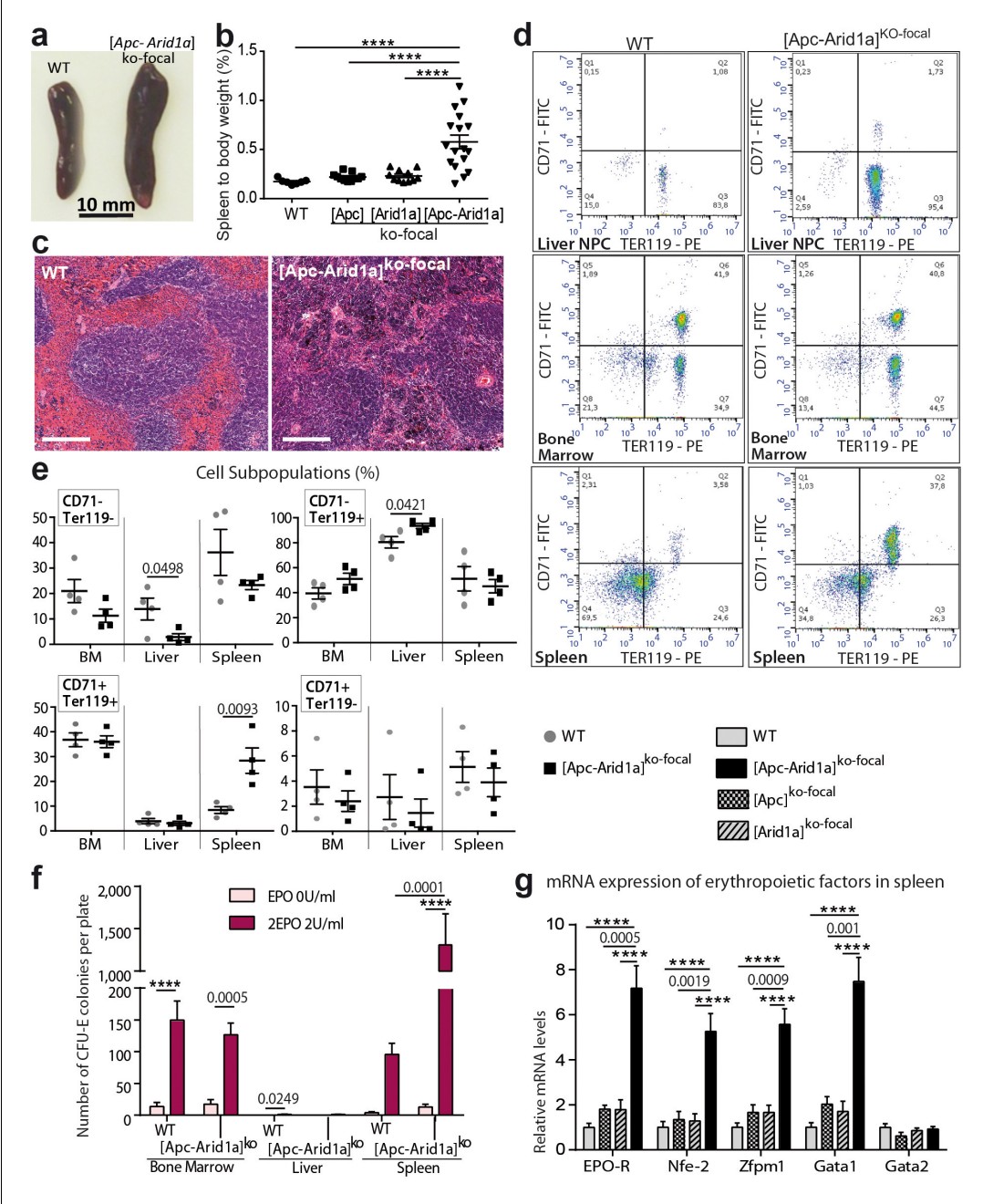

**Figure 3.** Erythropoiesis occurs in the spleen of [*Apc-Arid1a*]<sup>ko-focal</sup> mice. (a) Gross morphology of spleens from representative control (WT) and [*Apc-Arid1a*]<sup>ko-focal</sup> mice; (b) Spleen/body weight ratio of WT (n = 7), [*Apc*]<sup>ko-focal</sup> (n = 11), [*Arid1a*]<sup>ko-focal</sup> (n = 11), and [*Apc-Arid1a*]<sup>ko-focal</sup> (n = 17) mice (one-way ANOVA). (c) Hematoxylin and Eosin staining of splenic sections. Scale bar is 200 μm. (d,e) FACS analysis of liver NPC, bone marrow, and spleens from control (WT) or [*Apc-Arid1a*]<sup>ko-focal</sup> mice using the erythroid markers CD71 and Ter119. (e) FACS quantification from WT (n = 4) and [*Apc-Arid1a*]<sup>ko-focal</sup> (n = 4) mice (multiple t-test). (f) Quantification of erythroid progenitors as erythroid colony-forming units (CFU-E) in the presence of EPO, using $2 \times 10^5$ cells from bone marrow or $2 \times 10^6$ cells from the liver and spleen of WT or [*Apc-Arid1a*]<sup>ko-focal</sup> mice (2-way ANOVA). (g) Q-PCR showing relative expression of several factors, known to be involved in stress-induced erythropoiesis, in the spleens of WT (n = 9), [*Apc*]<sup>ko-focal</sup> (n = 5), [*Arid1a*]<sup>ko-focal</sup> (n = 8), and [*Apc-Arid1a*]<sup>ko-focal</sup> (n = 8) mice (one-way ANOVA). ****p<0.0001. Related data are found in *Figure 3—figure supplements 1* and source data in '*Figure 3—source data 1*'.

The online version of this article includes the following source data and figure supplement(s) for figure 3:

**Source data 1.** Spleen to body weight (*Figure 3b*), FACS analyses (*Figure 3e*), CFU-E counts (*Figure 3f*) and gene expression (*Figure 3g*).

**Figure supplement 1.** Hepato-specific and focal inactivation of *Apc* and *Arid1a* genes leads to sequestration of enucleated beta-globin-positive red blood cells.

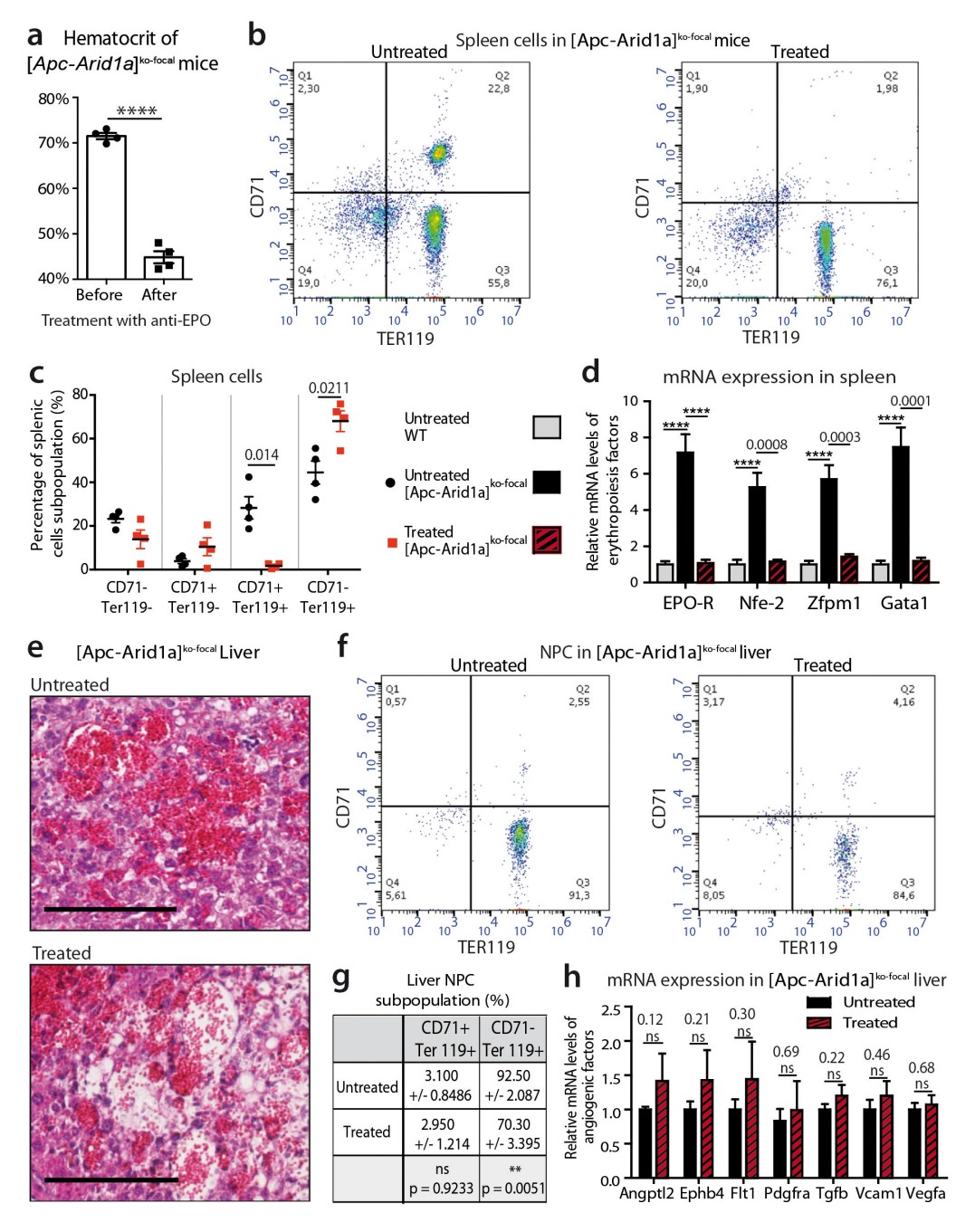

**Figure 4.** Blockade of Epo signaling with anti-EPO serum in [*Apc-Arid1a*]^ko-focal^ mice eliminates aberrant erythropoiesis in the spleen, but maintains angiogenesis in the liver. (**a**) Hematocrit before (n = 4) and after (n = 4) anti-EPO treatment (t-test). (**b,c**) FACS analysis (**b**) and quantification (**c**) of spleens with/without anti-EPO (n = 4 for each group) (t-test). (**d**) RT-qPCR showing relative expression of erythropoiesis factors in the spleens of WT (n = 9), treated [*Apc-Arid1a*]^ko-focal^ (n = 4), untreated [*Apc-Arid1a*]^ko-focal^ (n = 8) mice (one-way ANOVA). (**e**) Hematoxylin Eosin (HE)-stained sections of livers from representative 7-month-old mice. (**f,g**) FACS analysis (**f**) and quantification (**g**) of liver NPC with/without anti-EPO. (**h**) RT-qPCR showing relative expression of angiogenic factors in the livers with (n = 4) and without (n = 10) anti-EPO (t-test). ****p<0.0001. Related data are found in *Figure 4—figure supplements 1* and source data in '*Figure 4—source data 1*'.

The online version of this article includes the following source data and figure supplement(s) for figure 4:

**Source data 1.** Hematocrit (*Figure 4a*), FACS quantifications (*Figure 4c, g*) and gene expression (*Figure 4d, h*) after anti-EPO treatment.

**Figure supplement 1.** Anti-EPO blocking serum treatment in [*Apc-Arid1a*]^ko-focal^ mice leads to decrease of intra-hepatic red blood cells accumulation.

this decrease in intrahepatic RBCs, we did not observe any change in the disruption of the liver vascular architecture as shown by both histological (*Figure 4e*, *Figure 4—figure supplements 1*) and gene expression analyses (*Figure 4h*).

We demonstrate here that high plasma Epo concentration is directly responsible for splenic erythropoiesis and erythrocytosis in [*Apc-Arid1a*]$^{ko\text{-}focal}$ mice. However, this cytokine alone is not responsible for alterations in liver angiogenesis.

## *Epo* is cell-autonomously expressed by β-catenin-activated *Arid1a*-null hepatocytes in both the mouse and in humans

We investigated whether *Epo* is expressed by hepatocytes after *Apc* and/or *Arid1a* hepato-specific inactivations. We generated Tamoxifen-induced mouse models (*Figure 5a*) with short-term panlobular gene inactivations (*Figure 5b*) and *Apc* loss-induced hepatomegaly (*Figure 5—figure supplements 1a*) as previously shown (*Buenrostro et al., 2015*). After diet-based Tamoxifen administration, the *Apc* and/or *Arid1a* genes were invalidated in approximately 90% of hepatocytes (*Figure 5—figure supplements 1b*). There was no gene invalidation in liver NPCs, thus highlighting the high purity of the NPC fraction (*Figure 5—figure supplements 2a*). We detected *Epo* mRNA expression only in the hepatocyte compartment and not in NPCs of [*Apc-Arid1a*]$^{ko\text{-}TOTAL}$ livers, whereas a slight decrease of Epo expression was seen in the kidney of [*Apc-Arid1a*]$^{ko\text{-}TOTAL}$ mice (*Figure 5c,d*).

To confirm the cell-autonomous expression of *Epo* in β-catenin-activated *Arid1a*-null hepatocytes, we performed RNA in situ hybridization for *Epo* with *Axin2* as a marker of β-catenin activation (*Figure 6*). *Epo* transcripts were not expressed in the livers, yet were abundant in rare interstitial renal cells of control mice (*Figure 6—figure supplements 1a*); this localization of *Epo* in the kidney has already been described (*Lacombe et al., 1988*). Conversely but as expected, we found *Axin2* mRNA transcripts in pericentral hepatocytes (*Benhamouche et al., 2006*). After *Apc* and *Arid1a* gene invalidation, we found a de novo expression of *Epo* in a subset of *Axin2*-expressing hepatocytes. In the long-term focal model, this expression was restricted to the areas of peliosis (*Figure 6a*). In the short-term panlobular model, rare *Axin2*-expressing hepatocytes also expressed single *Epo* mRNA transcripts (*Figure 6b*). In both models, *Epo* expression was not found elsewhere in the liver.

We examined whether *Epo* expression is specific to the loss of *Apc* or can be initiated regardless of how Wnt/β-catenin signaling is activated. We successfully activated β-catenin via its Wnt/Spondin ligand in murine primary hepatocytes (*Figure 5—figure supplements 2e*). We consecutively performed in vivo Arid1a knockout followed by in vitro Wnt/Spondin stimulation, or in vivo Apc loss followed by efficient in vitro siRNA-mediated *Arid1a* knockdown (si-Arid1a) (*Figure 5—figure supplements 2f*). *Epo* expression significantly increased in these conditions (*Figure 5e*, *Figure 5—figure supplements 2b, c*). Mutational activation of β-catenin coupled with si-*Arid1a* also led to the induction of *Epo* expression in the β-catenin-mutated HEPA1.6 murine hepatoma-derived cell line (*Figure 5—figure supplements 2d*).

We assessed the conservation of EPO regulation from mouse to humans. We found that *EPO* mRNA expression was also regulated by both the chromatin remodeler ARID1A and the Wnt/β-catenin signaling pathway in primary human hepatocytes after siRNA-mediated *ARID1A* and *APC* downregulation (*Figure 5f*).

Overall, these in vivo and in vitro findings strongly demonstrate a conserved and cell-autonomous role of Wnt/β-catenin activation and *Arid1a* loss in hepatic *Epo* expression. This occurs as a stochastic transcriptional event in β-catenin-activated *Arid1a*-null hepatocytes.

## Wnt/β-catenin pathway control of 3' *Epo* enhancer activity is hypoxia- and HIF-independent

We questioned if β-catenin directly controls *Epo* transcription through cis-regulatory sequences. We previously performed ChIP-Seq experiments to assess Tcf4/β-catenin occupancy in the chromatin of hepatocytes isolated from [*Apc*]$^{ko\text{-}TOTAL}$ *versus* [*β-catenin*]$^{ko\text{-}TOTAL}$ murine models (*Gougelet et al., 2014*). The only DNA region bound by Tcf4 in the vicinity of the *Epo* gene was its 3'enhancer (*Epo*-3'E), known to be involved in *Epo* transcription in the embryonic liver, as well as the known Hif- (HIF-REs) and Hnf4-containing responses elements (HREs) (*Suzuki et al., 2011*; *Semenza et al., 1991*; *Figure 7a*). This Tcf4 binding was at the same location as HRE binding, and was stronger in activated

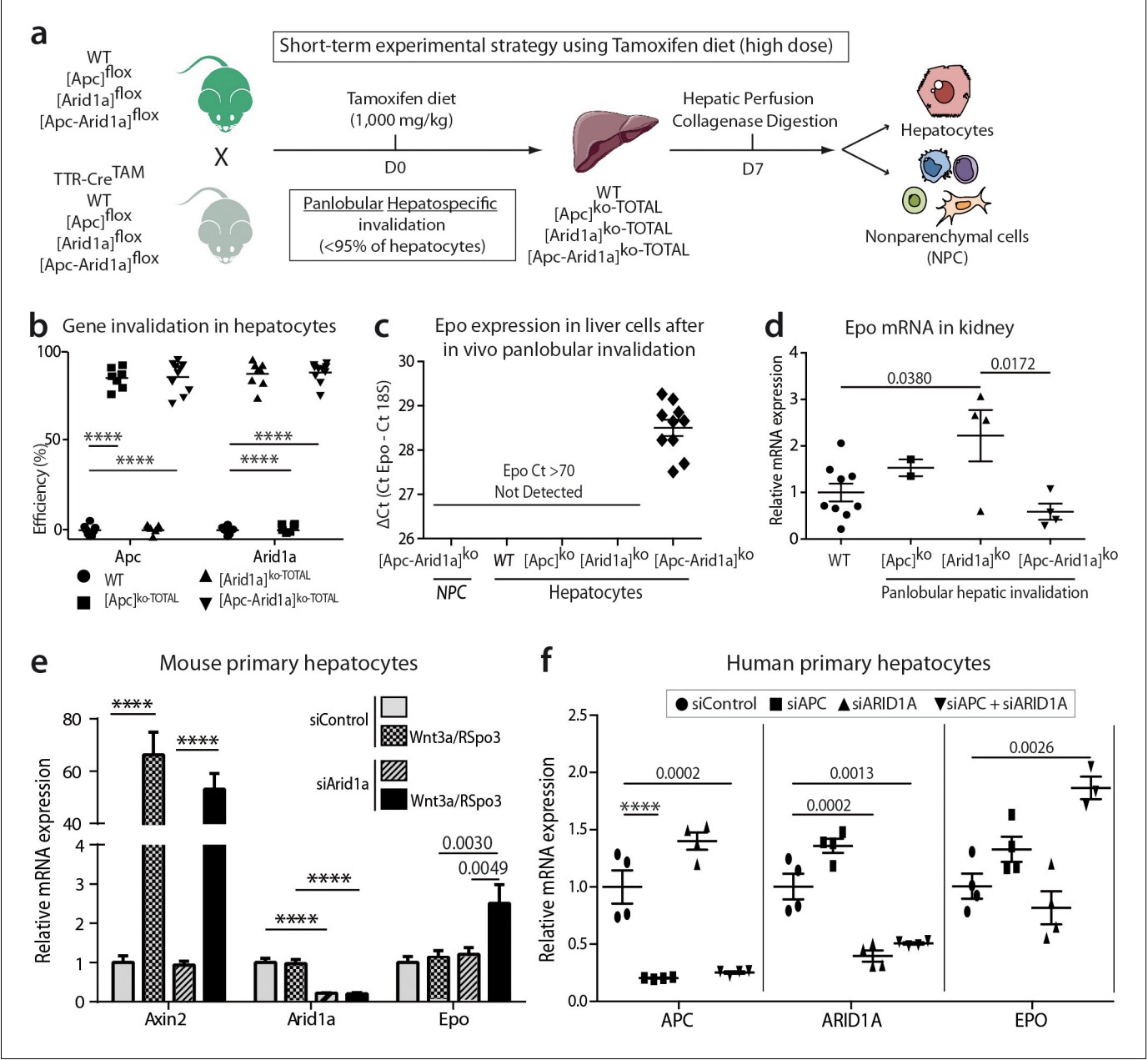

**Figure 5.** Cell-autonomous *Epo* expression after *Arid1a* inactivation and Wnt/β-catenin activation in murine and human hepatocytes. (**a**) In vivo and ex vivo strategy. WT (n = 8), [*Apc*]$^{ko\text{-}TOTAL}$ (n = 7), [*Arid1a*]$^{ko\text{-}TOTAL}$ (n = 8), and [*Apc-Arid1a*]$^{ko\text{-}TOTAL}$ (n = 10) mice. (**b**) Inactivation efficiency of *Apc* and *Arid1a* genes in isolated hepatocytes. (**c,d**) RT-qPCR assessment of erythropoietin (*Epo*) transcription (**c**) in the hepatocyte and NPC compartments of the livers, (**d**) in the kidney (1-way ANOVA). (**e**) In vitro analysis of *Axin2*, *Arid1a* (*Arid1a* floxed-exon detection), and *Epo* expression by RT-qPCR of mouse hepatocytes after Wnt3a and R-Spondin3 stimulation, and si-Arid1a/si-Control treatments, showing *Arid1a* knockdown efficiency and Wnt/β-catenin pathway activation, as the mRNA levels of *Axin2*, a canonical target gene of Wnt signaling, significantly increased (2-way ANOVA). (**f**) In vitro analysis of *Apc*, *Arid1a*, and *Epo* by RT-qPCR of cryopreserved human hepatocytes after siRNA transfection (one-way ANOVA analysis). Data are presented as the mean ± SEM. ****p<0.0001. Cell culture data are representative of three independent experiments. Related data are found in *Figure 5—figure supplements 1–2*, and source data in '*Figure 5—source data 1*; *Figure 5—figure supplement 1—source data 1*; *Figure 5—figure supplement 2—source data 1*'.

The online version of this article includes the following source data and figure supplement(s) for figure 5:

**Source data 1.** Efficiency of gene invalidation (*Figure 5b*), and gene expression in vivo and ex vivo (*Figure 5c-f*) in mice and humans.
**Figure supplement 1.** Panlobular inactivation of *Apc* and/or *Arid1a* in hepatocytes.
**Figure supplement 1—source data 1.** Liver to body weight ratio (*Figure 5—figure supplements 1a*).
**Figure supplement 2.** Cell-autonomous *Epo* expression after *Arid1a* invalidation and Wnt/β-catenin activation in hepatocytes.
*Figure 5 continued on next page*

*Figure 5 continued*

**Figure supplement 2—source data 1.** Efficiency of gene invalidation (*Figure 5—figure supplements 2a*), mRNA expression (*Figure 5—figure supplements 2b-d*), western blots (*Figure 5—figure supplements 2e*).

β-catenin than in β-catenin-null hepatocytes (*Figure 7a*). We demonstrated from ENCODE data that H3K27Ac, a histone mark indicating active promoters or enhancers, also bound to this region; this binding was present in mouse liver chromatin at E14.5, an embryonic stage in which the *Epo* gene is actively transcribed (*Figure 7a*). However, *Epo* was only partially present in the livers of eight-week-old mice, with no *Epo* transcription, and completely absent in the adult small intestine, a tissue known not to transcribe the *Epo* gene (*Figure 7a*).

We thus tested whether Wnt/β-catenin signaling directly activates hepatic *Epo* transcription through the *Epo*-3'E. We transfected a luciferase reporter (pEpoE-luc) containing the HIF and HNF4-binding sites into primary mouse hepatocytes (*Figure 7b*). After Wnt/Spondin stimulation, and regardless of si-*Arid1a* treatment, *Epo* enhancer activity was five- to eight-fold higher (*Figure 7c,d*). Hence, in this in vitro reporter assay context, β-catenin signaling increases *Epo*-3'E activity and it is independent of the chromatin landscape.

Hypoxia-inducible factor (HIF) signaling is the master pathway regulating *EPO* transcription and Hif2α has a prominent role in hepatic *Epo* transcription (*Mastrogiannaki et al., 2012*). We investigated Hif2α involvement in β-catenin/Arid1a-dependent *Epo* expression. In vivo, we did not detect hypoxia or Hif1α/Hif2α accumulation in the absence of *Apc* and/or *Arid1a* in mouse livers (*Figure 7—figure supplements 1a-c*). A small subset of Hif1α/Hif2α targets, such as Eno2, Car9, and Rab42, was slightly overexpressed in both [Apc]$^{ko}$ and [Apc-Arid1a]$^{ko}$ livers, confirming that β-catenin and HIF signaling share some transcriptional targets (*Figure 7—figure supplements 1d-e*; *Benhamouche et al., 2006*). As expected, the hypoxia-mimetic agent desferrioxamine (DFO) markedly potentiated luciferase activity in pEpoE-luc-transfected hepatocytes, whereas efficient knockdown of both *Hif1α* or *Hif2α* (*Figure 7—figure supplements 2*) resulted in a significant decrease (*Figure 7e*). Interestingly, knockdown of *HIFs*, either alone or combined, did not reduce *Epo*-3'E induction by β-catenin signaling in hepatocytes, whether *Apc* be inactivated alone or in combination with *Arid1a* (*Figure 7e*).

In all, the Wnt/β-catenin pathway controls erythropoietin expression in hepatocytes through the 3' *Epo* enhancer in a hypoxia- and HIF-independent manner.

## Both β-catenin signaling and Arid1a are key players in chromatin remodeling, histone recruitment, and Tcf4 binding on the hepatic *Epo* enhancer

We previously showed similarities between HREs and WREs, and that Tcf4 can bind HREs and thereby participate in β-catenin-dependent transcription (*Gougelet et al., 2014*). Here, we found that Tcf4 bound DNA on the HRE region of the Epo-3'E in which there is no classical WRE. Indeed, by electrophoretic mobility shift assay (EMSA), we showed that Tcf4 weakly bound the *Epo*-3'E HRE (thereafter called DR2) in control liver nuclear extracts (*Figure 8a*). In [*Apc*]$^{ko-TOTAL}$ liver extracts, the nuclear translocation of β-catenin led to a stronger binding represented by a supershift (*Figure 8a, b*). This indicates that the Tcf4/β-catenin complex binds this DR2 motif, as well as a classical WRE shown by competitive EMSA (*Figure 8b,c*). These findings highlighted that Tcf4 binds to the HRE of the *Epo* enhancer and that activation of β-catenin increases this interaction.

Endogenous hepatic *Epo* was expressed de novo after both Wnt/β-catenin activation and *Arid1a* knockout, but gene expression of classical β-catenin target genes (*Glul*, *Axin2*) was not affected by Arid1a status (*Figure 8—figure supplements 1*). We thus characterized Tcf4 binding, chromatin accessibility, and histone active (H3K27Ac) or repressive (H3K27Me3) marks of the Epo enhancer, the Axin2 intronic enhancer, and the Glul promoter in hepatocytes isolated from transgenic mouse livers.

As previously described (*Gougelet et al., 2014*), Tcf4 efficiently bound to the *Axin2* intronic enhancer in vivo and this increased when β-catenin signaling was activated (*Figure 8d*). This was

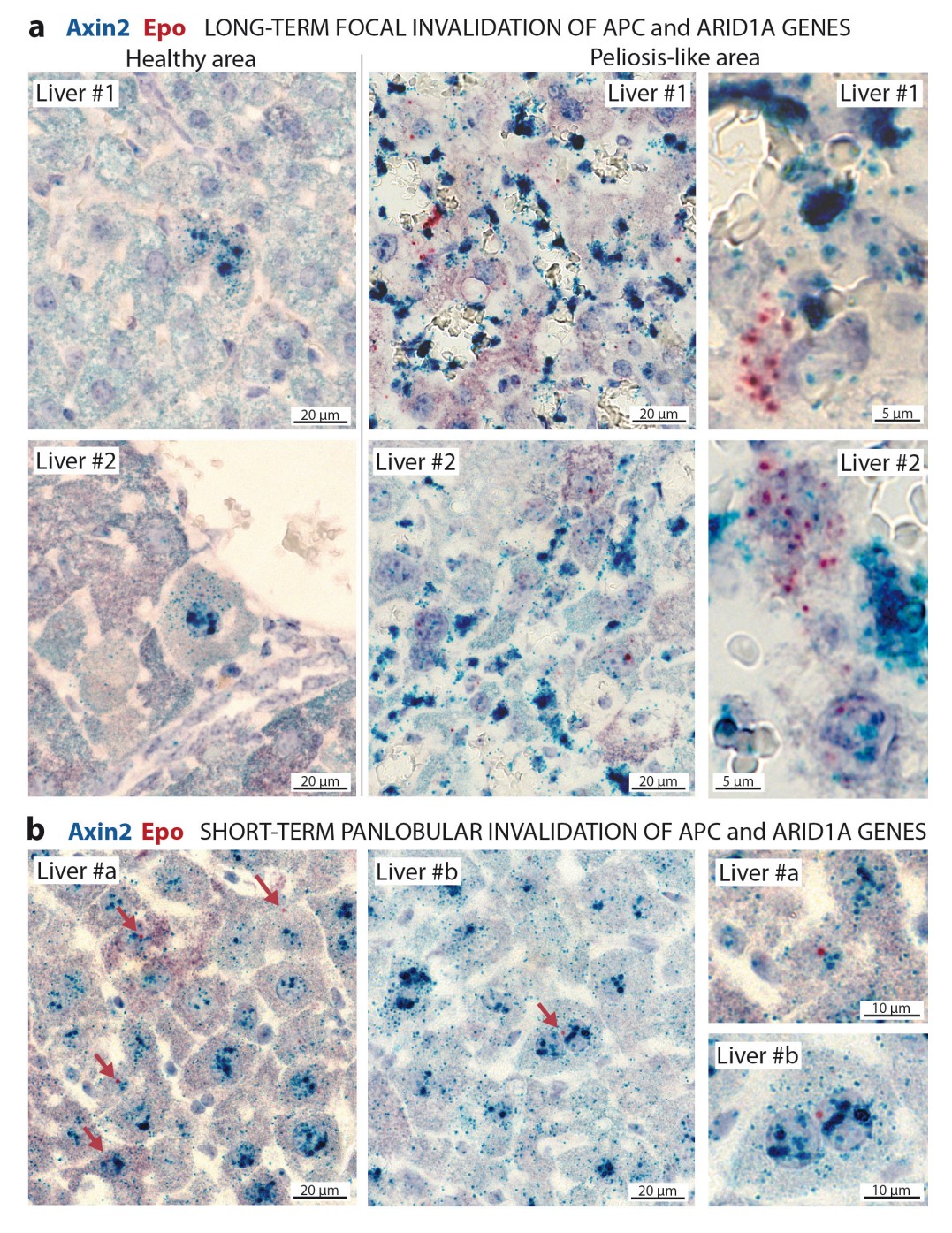

**Figure 6.** In situ hybridization of mRNAs showing a de novo expression of Epo in a subset of β-catenin-activated hepatocytes. (a) Seven months after Apc/Arid1a gene invalidation in single hepatocytes from two livers (#1 and #2); (b) 7 days after gene invalidation in more than 90% hepatocytes (two livers: #a and #b). Axin2 RNAScope probe stains β-catenin-activated hepatocytes (blue dots), and Epo RNAScope probe stains single Epo mRNAs as red dots. Related data are found in *Figure 6—figure supplement 1*.

The online version of this article includes the following figure supplement(s) for figure 6:

**Figure supplement 1.** Implementation of in situ Hybridization for Axin2 and Epo mRNAs using RNAScope, showing expressing mRNA as dots.

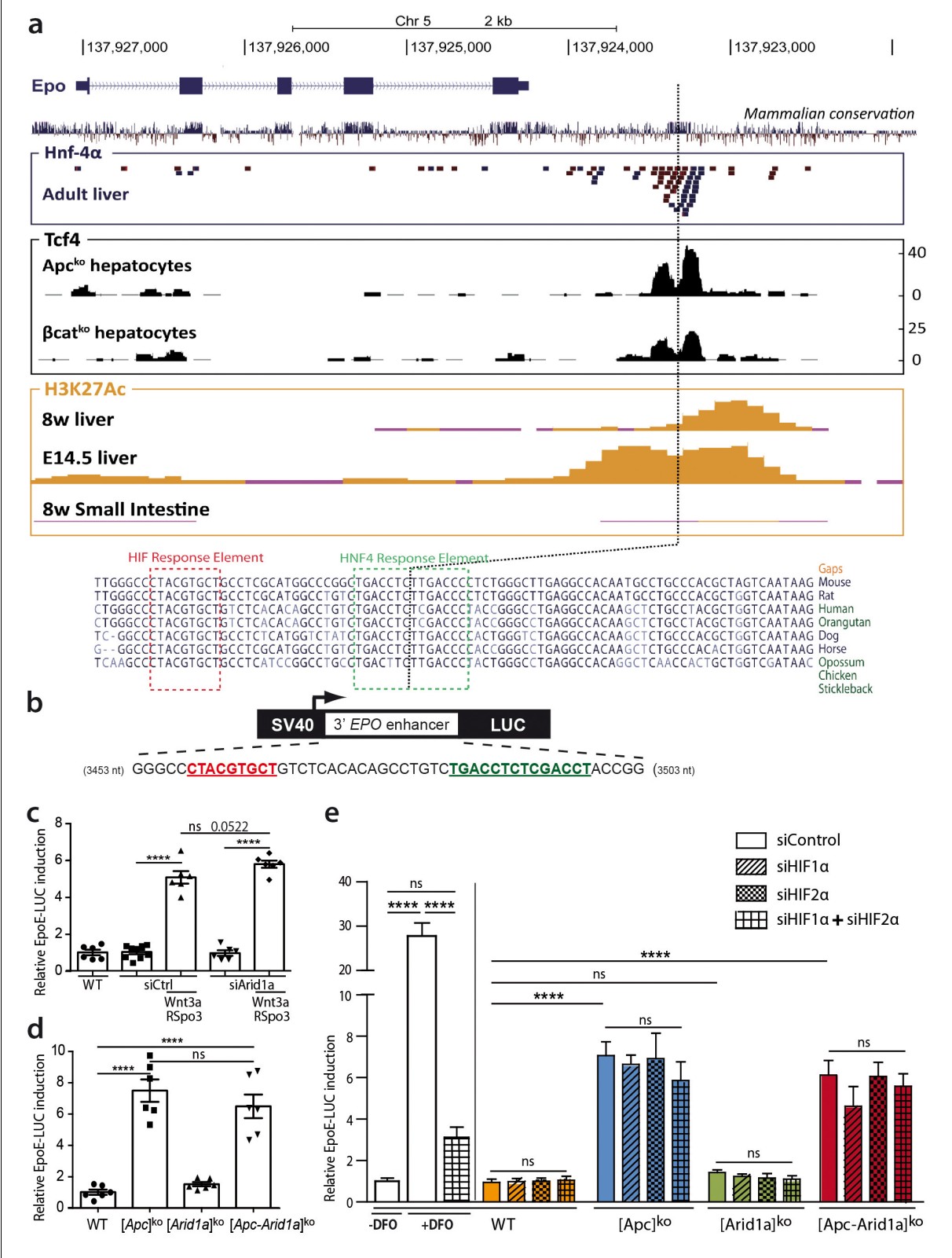

**Figure 7.** Wnt/β-catenin directly controls EPO expression through 3' *Epo* enhancer, in a HIF-independent manner. (**a**) Genomic environment of the *Epo* gene (UCSC Genome Browser, mm9 database) and ChIP-seq peaks at the 3' *Epo* enhancer. In blue/red: the crude reads of ChIP-Seq data performed in adult livers against HNF-4a (54). In black: ChIP-Seq under Apc[ko] or βcat[ko] conditions with an antibody against TCF4 (16). In yellow: ENCODE data of H3K27Ac marks in eight-week-old and E14.5 embryonic livers (Histone Mods by ChIP-Seq from ENCODE/LICR). (**b**) Schematic representation of the

*Figure 7 continued on next page*

*Figure 7 continued*

EpoE-Luc erythropoietin luciferase reporter, driven by the 3' enhancer. (**c–e**) Luciferase reporter assays in mouse primary hepatocytes: (**c**) after in vitro overactivation of Wnt/β-catenin signaling and Arid1a knockdown (**d**) after in vivo Cre-loxP-mediated gene inactivation; (**e**) Effect of hypoxic-mimic conditions using desferrioxamine (DFO), and effect of knockdown of HIF factors (two separate experiments carried out in triplicate). Results are in relative light units, and analyzed using 1-way (**d**) or 2-way ANOVA (**c,e**). ****p<0.0001. Related data are found in *Figure 7—figure supplements 1–2*, and source data in '*Figure 7—source data 1*; *Figure 7—figure supplement 1—source data 1*; *Figure 7—figure supplement 2—source data 1*'. The online version of this article includes the following source data and figure supplement(s) for figure 7:

**Source data 1.** EpoE-luc luciferase relative activity (*Figure 7c-e*).
**Figure supplement 1.** Lack of hypoxia and HIF signaling in [*Apc-Arid1a*]^ko-TOTAL livers.
**Figure supplement 1—source data 1.** Quantification of western blots (*Figure 7—figure supplements 1c*) and mRNA expression (*Figure 7—figure supplements 1d-e*).
**Figure supplement 2.** Effect of HIF1α and HIF2α knock-downs in mouse primary and transgenic hepatocytes.
**Figure supplement 2—source data 1.** mRNA expressions (*Figure 7—figure supplements 2a, c, d*) and western blots (*Figure 7—figure supplements 2b, e*).

correlated with a partial removal of the repressive H3K27Me3 mark (*Figure 8d*) and an increase in chromatin accessibility revealed by ATAC-qPCR analysis (*Figure 8f*). Co-inactivation of *Arid1a* and *Apc* decreased chromatin accessibility on this enhancer and induced a H3K27me3 repressive histone mark. A similar chromatin accessibility profile was seen for the Glul promoter. Tcf4 bound in vivo to the *Epo* enhancer, and this binding was slightly higher in [*Apc*]^ko-TOTAL and much higher in [*Apc-Arid1a*]^ko-TOTAL hepatocytes versus controls (*Figure 8e*). After *Apc* loss, the H3K27me3 repressive mark slightly decreased on *Epo* enhancer and chromatin was more accessible (*Figure 8f*, *Figure 8—figure supplements 1b*). In contrast, the loss of Arid1a strongly decreased the H3K27Me3 repressive mark without modifying chromatin access. In [*Apc-Arid1a*]^ko-TOTAL hepatocytes, the H3K27Ac active histone mark was induced while chromatin accessibility was lower compared to single knockout hepatocytes.

These data show that nuclear β-catenin favors Tcf4 binding on the *Epo* enhancer, increasing its chromatin accessibility, whereas *Arid1a* loss rather disrupts the H3K27me3 histone repressive mark. Both these changes increase the H3K27Ac enhancer mark and promote hepatic *Epo* transcription (*Figure 9*).

## Discussion

Our study shows that the Arid1a-dependent epigenetic landscape in the adult liver is a potent brake for transcription of EPO, a new key β-catenin target (*Figure 9*). Consequently, *Arid1a* loss in the context of β-catenin activation leads to Epo-dependent erythropoiesis in the spleen, erythrocytosis in the blood and liver, and to increased but defective angiogenesis, generating 'peliosis'.

Liver peliosis is a misunderstood human vascular disease, with non-specific features of impaired blood inflow and/or systemic inflammatory response (*Valla and Cazals-Hatem, 2018*). The dramatic phenotype we observed here is distinct from other existing murine models of liver-induced hypoxia with equivalent non-lethal erythrocytosis (*Minamishima and Kaelin, 2010*; *Ruschitzka et al., 2000*; *Takeda et al., 2008*). Using an anti-Epo blocking strategy, we could explain this discrepancy: our phenotypic observations were attributable to not only Epo-dependent erythrocytosis, as restricting plasma Epo rescued the erythrocytosis phenotype, but also to Epo-independent aberrant angiogenesis, a hallmark of liver peliosis (*Valla and Cazals-Hatem, 2018*).

We describe emerging roles for ARID1A and β-catenin signalings in *Epo* transcription. Until now, the major known regulator of both renal and hepatic *Epo* transcription was hypoxia-inducible factor signaling, acting via EPO's 3' enhancer (Epo-3'E) in the embryonic liver or anemic/hypoxic adult liver. We demonstrated that Tcf4/β-catenin bound the Hnf4-Responsive Element (Hnf4-RE) in the Epo-3'E and that the enhancer activity is independent of HIF in this context, contrary to what is reported in colorectal cancer cell lines, in which transcriptional cooperation between HIF and β-catenin occurs in hypoxia adaptation (*Kaidi et al., 2007*).

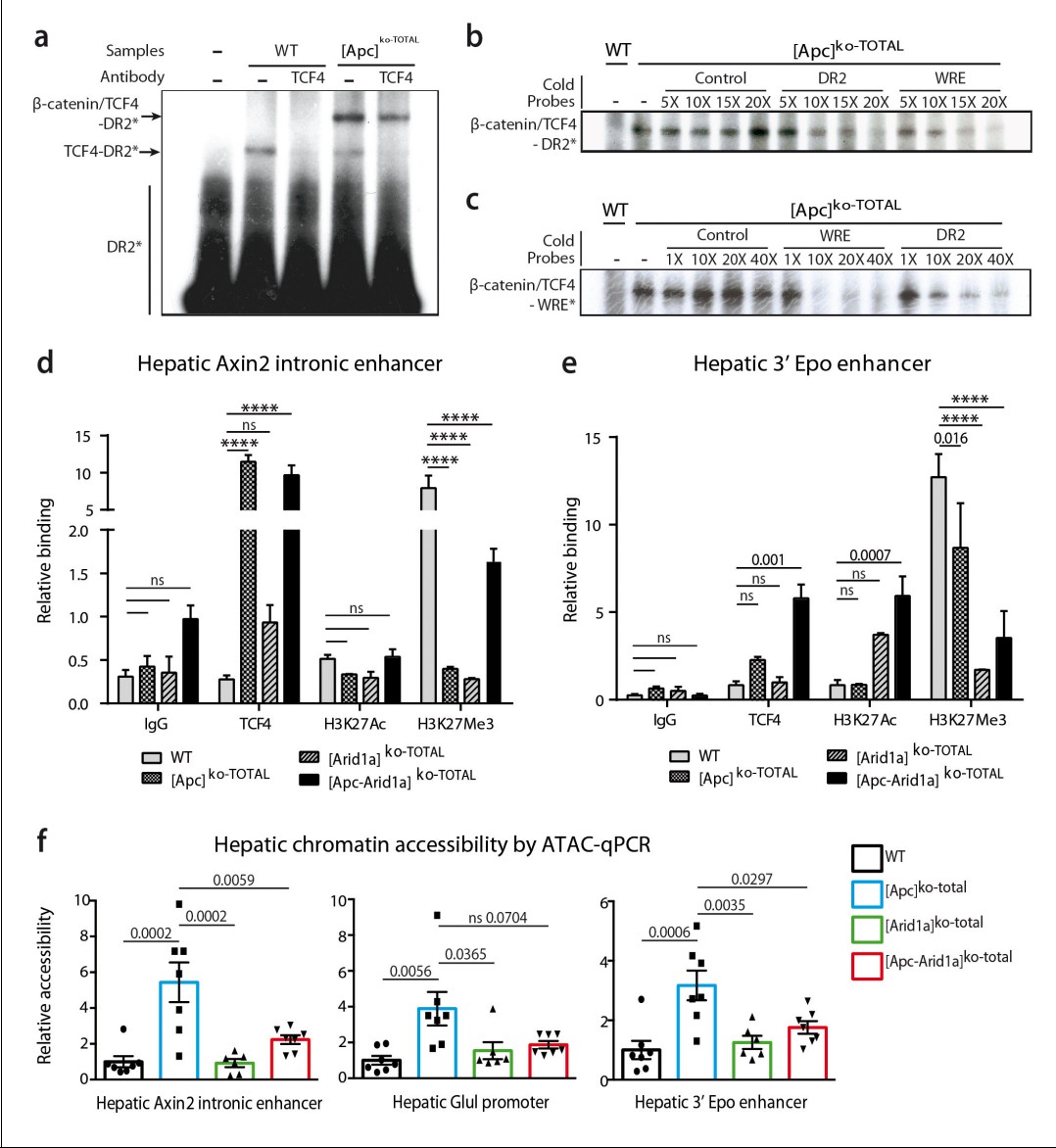

**Figure 8.** β-catenin/Tcf4 complex binds to the HNF4-responsive element of Epo enhancer (Epo-HRE) after modifications of histone marks and chromatin accessibility. (a) EMSA using nuclear proteic extracts from WT or [Apc]ko-TOTAL livers and 32P-labeled probes containing Epo-HRE (DR2). (b, c) Competitive EMSA using 32P-labeled DR2 (b) and 32P-labeled WRE (c) probes and increasing concentrations of cold probes containing HNF4, WRE or control-responsive element. WRE cold probes compete with radiolabeled DR2 motif for the Tcf4/β-catenin binding and vice versa. (d, e) Chromatin ImmunoPrecipitation (ChIP) assays of hepatocytes from WT, [Apc]ko-TOTAL, [Arid1a]ko-TOTAL, and [Apc-Arid1a]ko-TOTAL livers. ChIP-qPCR against IgG, Tcf4, Acetylation of Histone3 in Lysine27 (H3K27Ac), and Tri-methylation of Histone3 in Lysine27 (H3K27me3) for *Axin2* (d) and *Epo* (e) enhancer regions. WT (n = 3), [Apc]ko-TOTAL (n = 2), [Arid1a]ko-TOTAL (n = 2), and [Apc-Arid1a]ko-TOTAL (n = 3) mice. Enrichment by ChIP was assessed relative to the input DNA and normalized to the level of negative controls. (f) ATAC-qPCR using frozen livers from WT (n = 7), [Apc]ko-TOTAL (n = 7), [Arid1a]ko-TOTAL (n = 6), and [Apc-Arid1a]ko-TOTAL (n = 7) mice. Data are analyzed with one-way ANOVA. ****p<0.0001. Related data are found in *Figure 8—figure supplements 1–2*, and source data in '*Figure 8—source data 1*; *Figure 8—figure supplement 2—source data 1*; *Figure 8—figure supplement 2— source data 1*'.

The online version of this article includes the following source data and figure supplement(s) for figure 8:

**Source data 1.** EMSA (*Figure 8a-c*), ChIP-qPCR (*Figure 8d, e*) and ATAC-qPCR (*Figure 8f*) data.

**Figure supplement 1.** The expression of β-catenin-positive target genes is not modulated by Arid1a status.

**Figure supplement 1—source data 1.** mRNA expression (*Figure 8—figure supplements 1a*).

**Figure supplement 2.** Chromatin accessibility assessed all along the hepatic 3'Epo enhancer by ATAC-qPCR.

**Figure supplement 2—source data 1.** ATAC-qPCR data (*Figure 8—figure supplements 2b*).

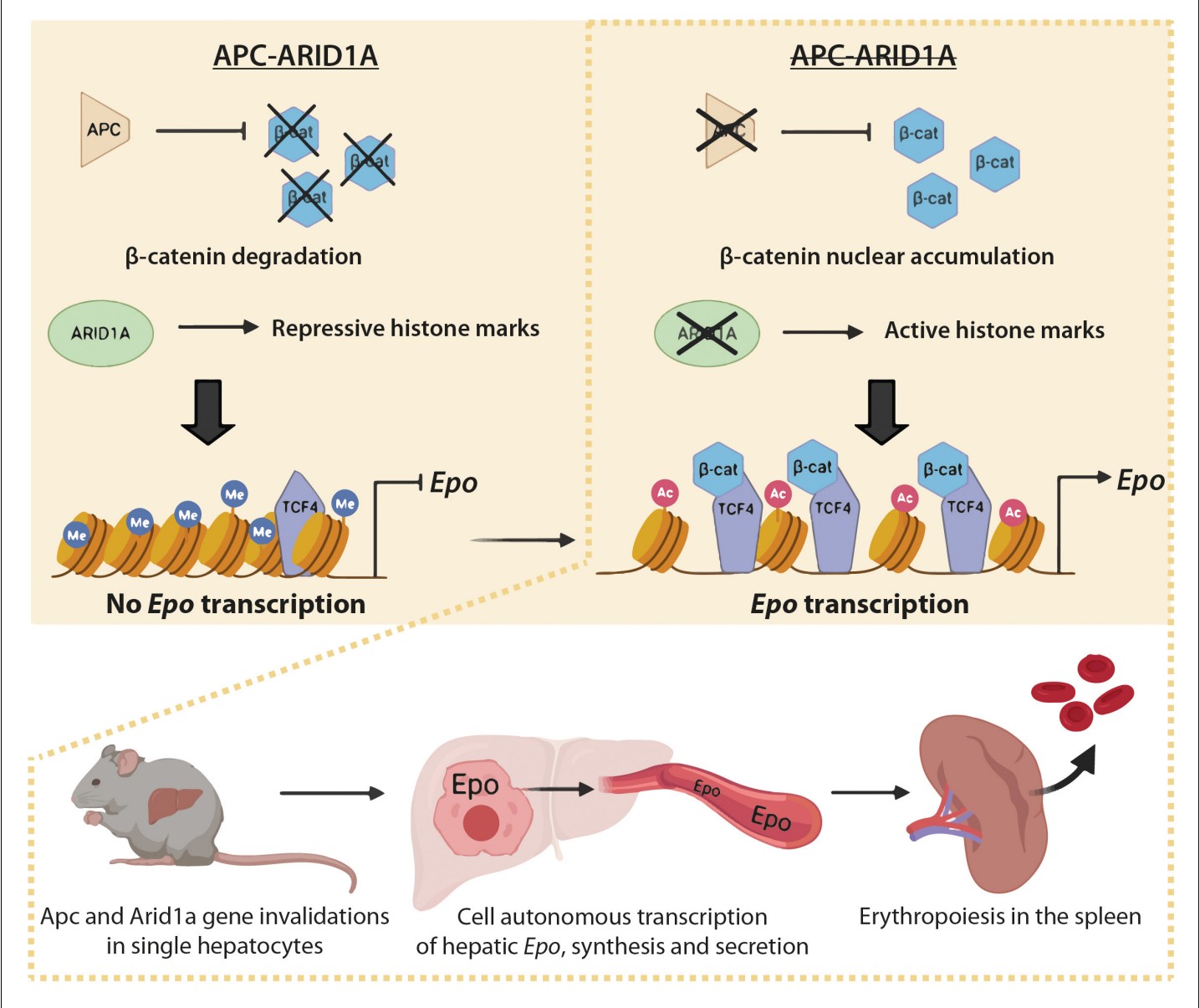

**Figure 9.** Schematic model of the role of *Arid1a* in hepatic *Epo* expression linked to overactivation of the Wnt/β-catenin pathway. Under physiological conditions, the presence of Arid1a is associated with histone repressive marks at the *Epo* enhancer and β-catenin is constantly degraded; thus, Epo is not produced. In the absence of Apc, β-catenin/Tcf4 complex binds the *Epo* enhancer, and enhances chromatin accessibility, but the histone marks remain repressive. The loss of *Arid1a* increases active histone marks, which is insufficient to induce *Epo* transcription. After both Wnt/β-catenin activation and Arid1a inactivation, active histone marks and binding of β-catenin/Tcf4 to the *Epo* enhancer drive *Epo* liver transcription, and subsequent secretion of Epo into the bloodstream, resulting in splenic erythropoiesis and in substantial blood and liver erythrocytosis.

The consequences of this HIF signaling-independent *Epo* regulation is significant for the genetic engineering of EPO for therapeutic purposes. In anemia, a major complication of chronic kidney disease, HIF stabilizers are currently used to restore circulating EPO levels. The long-term safety of this strategy is hindered by the lack of targeting specificity (*Kular and Macdougall, 2019*). The use of cell transcriptional machinery to produce therapeutic levels of EPO has been put forward to overcome the side effects associated with HIF stabilizers. The EPO-producing cells of the adult kidney are potential candidates, but anemic patients have damaged kidneys. Based on our results, here we

can propose an alternative involving the restoration of the ability of hepatocytes to synthetize EPO, independently of hypoxia, by targeting Wnt/β-catenin and ARID1A signaling in the liver.

Our demonstration that *Arid1a* inactivation is required in *Epo* transcription opposes previously described roles of chromatin remodeling complexes in hepatic regulation of *Epo* (*Wang et al., 2004*; *Sena et al., 2013*). However, firstly these studies analyzed hypoxia-dependent Epo regulation which is distinct from our study; we firmly established that the β-catenin-dependent control of *Epo* transcription depends on *Arid1a* loss, is Hif-independent, and occurs in a normoxic adult liver. Secondly, these studies focused on BRG1/BRM ATPases, essential core subunits of both the BAF and pBAF complexes. The loss of *Arid1a*, a facultative component of the BAF complex, does not disrupt BAF complex functionality as Arid1b is known to compensate for Arid1a loss. This highlights a specific role for Arid1a in transcriptional repression through the modulation of chromatin accessibility to transcription factors at their target DNA sequences (*Sun et al., 2016*; *Nagl et al., 2005*). We show increased binding of the Tcf4/β-catenin complex to Epo-3'E Hnf4-RE is *Arid1a*-dependent and *Arid1a* loss decreases the H3K27me3 repressive mark. That could be due to the intricate balance existing between the Polycomb complex PRC2 and the SWI/SNF complex (*Kadoch et al., 2016*). Accordingly, the inhibition of the Polycomb EZH2 subunit is synthetically lethal in ARID1A-mutated tumors (*Bitler et al., 2015*; *Alldredge and Eskander, 2017*). Therefore, Arid1a and the Polycomb complex could act in concert to modulate *Epo* gene expression in the liver.

We illustrate that *Arid1a* loss renders the liver *Epo*-inducible element more accessible to Tcf4, and even more so to β-catenin. Contrary to the paradigm that Tcf4 binds its DNA targets regardless of β-catenin activation status, we previously reported that Tcf4 DNA-binding was stronger in the presence of nuclear β-catenin in the liver (*Gougelet et al., 2014*). More broadly, numerous interactions between chromatin remodeling and Wnt/β-catenin signaling have already been described (*Barker et al., 2001*; *Eckey et al., 2012*; *Mathur et al., 2017*; *Song et al., 2009*; *Yan et al., 2014*; *Zhai et al., 2016*) and can explain the impact of β-catenin signaling on chromatin accessibility at the *Epo* enhancer. Single-RNA in situ hybridization revealed that *Epo* gene expression only occurs in rare hepatocytes, emphasizing the complexity of Epo liver transcription in the liver. This contributes to previous studies using single-RNA in situ hybridization, showing that transcription in the liver is gene-dependent, and is either bursty and dynamic or stable (*Bahar Halpern et al., 2015*).

We found here that the loss of Arid1a does not change the transcription of hepatic canonical Wnt/β-catenin target genes. As for *Epo*, it could potentially unmask new chromatin-dependent β-catenin target genes. Among these new Arid1a/β-catenin target genes are those involved in liver angiogenesis. In the near future, genome-wide studies will be required to firmly identify these genes, combining the analysis of transcriptome, chromatin accessibility (ATAC-Seq), histone mark, β-catenin and Arid1a cistromes (ChIP-Seq) in liver chromatin from Arid1a-null and β-catenin-activated hepatocytes.

The initial aim of our study was to better elucidate oncogenic cooperation in liver carcinogenesis. In our in vivo experimental models reported here, the loss of Arid1a protects against β-catenin-dependent carcinogenesis. However, these results were not fully exploitable due to the deleterious effect of the dramatic hematological disorder developed by the mice. New mouse models are therefore required for further investigation of the oncogenic role of *Arid1a* in liver carcinogenesis. In turn, confirmation of such a role would corroborate a recent study showing that hepatic *Arid1a* can harbor either a tumor suppressor or oncogenic role depending on the cellular context (*Sun et al., 2018*). An additional study also demonstrated that *Arid1a* is protumoral rather than a tumor suppressor in colorectal cancer with *Apc* mutations (*Mathur et al., 2017*).

Lastly, some liver cancer studies have identified pathological erythrocytosis and/or hepatic vascular lesions, potentially with EPO production and peliosis. However, the molecular mechanisms underlying these pathological observations are still poorly understood (*Matsuyama et al., 2000*; *Bunn, 2013*; *Ke et al., 2017*; *Hoshimoto et al., 2009*; *Tsuchiya et al., 2009*; *Vik et al., 2009*). Our study contributes molecular clues by indicating that this is not linked to *CTNNB1/ARID1A* mutations, but more likely attributed to the hypoxia frequently found in cancers. Future studies should use mouse models and data from patients with HCC to address the specific transcriptional output of *CTNNB1/ARID1A*-mutated liver tumors.

# Materials and methods

**Key resources table**

| Reagent type (species) or resource | Designation | Source or reference | Identifiers | Additional information |
|---|---|---|---|---|
| Gene (*Mus musculus*) | Epo | GenBank | NM_007942.2 | Erythropoietin |
| Gene (*Mus musculus*) | Arid1a | GenBank | NM_001080819.2 | Arid1a |
| Gene (*Mus musculus*) | Ctnnb1 | GenBank | NM_007614.3 | Beta-catenin |
| Gene (*Mus musculus*) | Apc | GenBank | NM_001360980.1 | Adenomatous polyposis coli |
| Strain, strain background (*Mus musculus*) | Arid1a-lox | From Z. Wang's lab | *Arid1a*$^{tm1.1Zhwa}$/J | https://www.jax.org/strain/027717 |
| Strain, strain background (*Mus musculus*) | Apc-lox | From Perret-Colnot's lab | Apc$^{tm2.1Cip}$ | https://www.infrafrontier.eu/search?keyword=EM:05566 |
| Strain, strain background (*Mus musculus*) | Ttr-Cre-Tam | From Perret-Colnot's lab | Tg(Ttr-cre/Esr1*)1Vco | https://www.infrafrontier.eu/search?keyword=EM:01713 |
| Genetic reagent (*Adenovirus 5*) | Ad-Cre | Université de Nantes, France | Ad5-CAG-Cre | https://umr1089.univ-nantes.fr/facilities-cores/cpv/translational-vector-core-2201753.kjsp?RH=1519296751975 |
| Cell line (*Mus musculus*) | Mouse hepatoma | From Christine Perret's lab | Hepa 1-6 [Hepa1-6] (ATCC CRL-1830) | For transfection experiments |
| Antibody | anti-Arid1a (Rabbit monoclonal) | Abcam | Cat# 182560 [EPR13501] | IHC(1:1000), WB (1:2000) |
| Antibody | anti-Glul (GS) (Mouse monoclonal) | BD Biosciences | Cat# 610518, RRID:AB_397880 | IHC(1:400), WB (1:5000) |
| Antibody | anti-HBB (Mouse monoclonal) | Proteintech | Cat# 16216–1-AP, RRID:AB_10598329 | IHC(1:200), WB (1:2000) |
| Antibody | anti-HIF1α (Rabbit polyclonal) | Novus | Cat# NB100-449, RRID:AB_10001045 | WB nuclear extract (1:500) |
| Antibody | anti-HIF2α (Rabbit polyclonal) | Novus | Cat# NB100-122, RRID:AB_10002593 | WB nuclear extract (1:500) |
| Antibody | Anti-Tcf4 (Tcf7l2) (Mouse monoclonal) | Millipore | Cat# 05–511, RRID:AB_309772 | ChIP: 3 µg |
| Antibody | Anti-H3K27Ac (Rabbit polyclonal) | Active Motif | Cat# 39133, RRID:AB_2561016 | ChIP: 3 µg |
| Antibody | Anti-H3K27me3 (Rabbit polyclonal) | Active Motif | Cat# 39155, RRID:AB_2561020 | ChIP: 3 µg |

*Continued on next page*

*Continued*

| Reagent type (species) or resource | Designation | Source or reference | Identifiers | Additional information |
|---|---|---|---|---|
| Antibody | IgG (Mouse) | Thermo Fisher Scientific | Cat# 10400C, RRID:AB_2532980 | ChIP: 3 µg |
| Antibody | Anti-CD71-FITC (Rat monoclonal) | BD Biosciences | Cat# 553266, RRID:AB_394743 | FACS (1:100) |
| Antibody | Anti-Ter119-PE (rat monoclonal) | BD Biosciences | Cat# 553673, RRID:AB_394986 | FACS (1:100) |
| Antibody | Anti-β-actin (mouse monoclonal) | Sigma-Aldrich | Cat# A5441, RRID:AB_476744 | WB (1:10000) |
| Antibody | Anti-lamin A/C (rabbit polyclonal) | Cell Signaling Technology | Cat# 2032, RRID:AB_2136278 | WB nuclear extract (1:500) |
| Antibody | IgG, HRP-conjugated (horse, anti-mouse) | Cell Signaling Technology | Cat# 7076, RRID:AB_330924 | WB (1:2000) |
| Antibody | IgG, HRP-conjugated (goat, anti-rabbit) | Cell Signaling Technology | Cat# 7074, RRID:AB_2099233 | WB (1:2000) |
| Antibody | IgG, biotinylated (goat, anti-rabbit) | Vector lab | Cat# BA-1000, RRID:AB_2313606 | IHC (1:200) |
| Commercial assay or kit | MOM mouse on mouse | Vector Laboratories | Cat# BMK-2202, RRID:AB_2336833 | Kit |
| Sequence-based reagent | 18S | Thermo Fisher Scientific | Taqman Assay 4308329 | qPCR primers |
| Sequence-based reagent | Glul | Thermo Fisher Scientific | Taqman Assay Mm00725701_si | qPCR primers *Mus musculus* |
| Sequence-based reagent | Axin2 | Thermo Fisher Scientific | Taqman Assay Mm00443610_m1 | qPCR primers *Mus musculus* |
| Sequence-based reagent | Arid1a (total) | Thermo Fisher Scientific | Taqman Assay Mm00473838_m1 | qPCR primers *Mus musculus* |
| Sequence-based reagent | Arid1a (not excised by Cre) | Thermo Fisher Scientific | Taqman Assay Mm00473841_m1 | qPCR primers *Mus musculus* |
| Sequence-based reagent | Apc (total) | Thermo Fisher Scientific | Taqman Assay Mm00545877_m1 | qPCR primers *Mus musculus* |
| Sequence-based reagent | Apc (not excised by Cre) | Thermo Fisher Scientific | Taqman Assay Mm01130462_m1 | qPCR primers *Mus musculus* |
| Sequence-based reagent | Epo | Thermo Fisher Scientific | Taqman Assay Mm01202755_m1 | qPCR primers *Mus musculus* |
| Sequence-based reagent | 18 s | Eurogentec | F_GTAACCCGT TGAACCCCATT R_CCATCCAA TCGGTAGCG | SybrGreen qPCR primers |
| Sequence-based reagent | Angiopoietin-like 2 (Angptl2) | Eurogentec | F_CCGCAACAT GAACTCGAGAG R_GTGCTCCAGG TCCTTGTACT | SybrGreen qPCR primers *Mus musculus* |
| Sequence-based reagent | Carbonic anhydrase 9 (Car9) | Eurogentec | F_GACCTCGTG ATTCTCGGCTA R_GAGAAGGC CAAACACCAAGG | SybrGreen qPCR primers *Mus musculus* |

*Continued on next page*

*Continued*

| Reagent type (species) or resource | Designation | Source or reference | Identifiers | Additional information |
|---|---|---|---|---|
| Sequence-based reagent | Cyclin D1 (Ccnd1) | Eurogentec | F_AGAAGTGCG AAGAGGAGGTC R_TTCTCGGC AGTCAAGGGAAT | SybrGreen qPCR primers *Mus musculus* |
| Sequence-based reagent | Enolase 2, gamma neuronal (Eno2) | Eurogentec | F_TGGATTTCA AGTCTCCCGCT R_TCAGGTCAT CGCCCACTATC | SybrGreen qPCR primers *Mus musculus* |
| Sequence-based reagent | Erythropoietin receptor (Epo-r) | Eurogentec | F_ATGACTTTCG TGACTCACCCT R_GGGCTCCG AAGAACTTCTGTG | SybrGreen qPCR primers *Mus musculus* |
| Sequence-based reagent | FMS-like tyrosine kinase 1 (Flt1) | Eurogentec | F_AGAGGAGGA TGAGGGTGTCT R_GGGAACTT CATCTGGGTCCA | SybrGreen qPCR primers *Mus musculus* |
| Sequence-based reagent | GATA binding protein 1 (Gata1) | Eurogentec | F_TTCCCACTA CTGCTGCTACC R_GCGGCCTC TATTTCAAGCTC | SybrGreen qPCR primers *Mus musculus* |
| Sequence-based reagent | GATA binding protein 2 (Gata2) | Eurogentec | F_GCCGGTTCT GTCCATTCATC R_ATGGCAGCA GTCTCTTCCAT | SybrGreen qPCR primers *Mus musculus* |
| Sequence-based reagent | Inhibin beta-B (Inhbb) | Eurogentec | F_GTACCTGAAA CTGCTCCCCT R_ATGGCCTC TGTGATGGGAAA | SybrGreen qPCR primers *Mus musculus* |
| Sequence-based reagent | Potassium channel tetramer domain contain. 11 (Kctd11) | Eurogentec | F_TGACTTCTAC CAGATCCGGC R_TCAGGGTCAG TGCAGAAGAG | SybrGreen qPCR primers *Mus musculus* |
| Sequence-based reagent | Kinase insert domain protein receptor (Kdr) | Eurogentec | F_AGAAGATGC CCATGACCCAA R_TCACCCATC CTCAACACACA | SybrGreen qPCR primers *Mus musculus* |
| Sequence-based reagent | Nuclear factor, erythroid derived 2 (Nfe2) | Eurogentec | F_GATGTCCCGA ACTAGAGCCA R_ACACCCTTG GCCTTAGAGTC | SybrGreen qPCR primers *Mus musculus* |
| Sequence-based reagent | Platelet derived growth factor receptor, alpha polypeptide (Pdgfra) | Eurogentec | F_ACAGCTCAC AGACTTCGGAA R_AGAAGATGA TACCCGGAGCG | SybrGreen qPCR primers *Mus musculus* |
| Sequence-based reagent | Phosphoglycerate kinase 1 (Pgk1) | Eurogentec | F_TGGCACCAG GAACCCTTAAA R_AGCTCAGCC TTTACAGCTCA | SybrGreen qPCR primers *Mus musculus* |
| Sequence-based reagent | Placenta-specific 8 (Plac8) | Eurogentec | F_TGATTGCTT CAGTGACTGCG R_GTTCATGGC TCTCCTCCTGT | SybrGreen qPCR primers *Mus musculus* |
| Sequence-based reagent | Protein tyrosine phosphatase, receptor type, B (Ptprb) | Eurogentec | F_TGGACCCTG GGATCTAAGGA R_GTGGTCACT GCAAGCTTCAA | SybrGreen qPCR primers *Mus musculus* |

*Continued on next page*

*Continued*

| Reagent type (species) or resource | Designation | Source or reference | Identifiers | Additional information |
|---|---|---|---|---|
| Sequence-based reagent | Member RAS oncogene family (Rab42) | Eurogentec | F_GGCGTTCTG TTGGTCTTTGA R_GCAAGTTCCT CTGCTTCCTG | SybrGreen qPCR primers *Mus musculus* |
| Sequence-based reagent | Vascular endothelial growth factor A (Vegfa) | Eurogentec | F_GCTGTAACGAT GAAGCCCTG R_CGCTCCAGG ATTTAAACCGG | SybrGreen qPCR primers *Mus musculus* |
| Sequence-based reagent | Zinc finger protein, multitype 1 (Zfpm1) | Eurogentec | F_CCTTGAGATG GCGTTCACAG R_CCTGCTCTA CTACTGTGCCA | SybrGreen qPCR primers *Mus musculus* |
| Sequence-based reagent | AT-rich interaction domain 1A (ARID1A) | Eurogentec | F_AAGCCACCAA CTCCAGCATCCA R_CGCTTCTGG AATGTGGAGTCAC | SybrGreen qPCR primers (*Homo sapiens*) |
| Sequence-based reagent | Adenomatous polyposis coli (APC) | Eurogentec | F_CACACTTCCAA CTTCTCGCAACG R_AGGCTGCAT GAGAGCACTTGTG | SybrGreen qPCR primers (*Homo sapiens*) |
| Sequence-based reagent | Erythropoietin (EPO) | Eurogentec | F_GCATGTGGAT AAAGCCGTCAGTG R_GAGTTTGCGGA AAGTGTCAGCAG | SybrGreen qPCR primers (*Homo sapiens*) |
| Sequence-based reagent | DOS7-binding site (Control) | Eurogentec | F_GGGGTAGG AACCAATGAAA R_TTTCATTGG TTCCTACCCC | EMSA probe *Mus musculus* |
| Sequence-based reagent | HNF4-responsive element (DR2) | Eurogentec | F_GCCCGGCTGACC TCTTGACCCCTCT GGGCTTGAG R_CTCAAGCCCAGA GGGGTCAAGAG GTCAGCCGGGC | EMSA probe *Mus musculus* |
| Sequence-based reagent | Wnt-reponsive element | Eurogentec | F_CATCCCCCT TTGATCTTACC R_GGTAAGATC AAAGGGGGATG | EMSA probe |
| Sequence-based reagent | Negative control region | Eurogentec | F_ACACACCTT GAATCCCGT R_CCCAGCTA GAATGAACAAG | qPCR primers for ChIP and ATAC |
| Sequence-based reagent | Hepatic Epo 3' enhancer | Eurogentec | F_CTGTACCTCA CCCCATCTGGTC R_CCCAGCTCA CTCAGCACTTGTCC | qPCR primers for ChIP and ATAC |
| Sequence-based reagent | EPO-enh-5' (1) | Eurogentec | F_GGCAACAGC TGAAATCACCAA R_TCCCAGATC TGATGCCTTGC | qPCR primers for ATAC |
| Sequence-based reagent | EPO-enhHIF (2) | Eurogentec | F_CTGTACCTC ACCCCATCTGG R_CAGAGGG GTCAAGAGGTCAG | qPCR primers for ChIP and ATAC |
| Sequence-based reagent | EPO-enhHnf4 (3) | Eurogentec | F_GCAAGGCAT CAGATCTGGGA R_AGACAGCCT TGAATGGAGCC | qPCR primers for ChIP and ATAC |

## Animals

Mice carrying two floxed alleles in the 14th exon of the *Apc* gene (generated in our laboratory [*Colnot et al., 2004*]) or the 8th exon of the *Arid1a* gene (created by the Zhong Wang laboratory [*Gao et al., 2008*]),were interbred with TTR-Cre$^{Tam}$ mice (*Tannour-Louet et al., 2002*), resulting in *Apc*$^{flox/+}$/TTR-Cre$^{Tam}$ or *Arid1a*$^{flox/+}$/TTR-Cre$^{Tam}$ mice. For focal genetic inactivation, 8-week-old *Apc*$^{flox/flox}$ and *Arid1a*$^{flox/flox}$ male mice were injected intravenously with $0.5 \times 10^9$ infectious particles of Ad5-CAG-cre (AdCre) adenovirus as described (*Colnot et al., 2004*). Mice with hepato-specific and AdCre-mediated inactivation of *Apc* and/or *Arid1a* in single hepatocytes are referred to as [*Apc-Arid1a*]$^{ko-focal}$, [*Apc*]$^{ko-focal}$, and [*Arid1a*]$^{ko-focal}$ mice. The development of tumors and peliosis were followed monthly by 2D-ultrasound (Vevo 770, Visualsonics). For panlobular genetic inactivation, 8-week-old *Apc*$^{flox/flox}$/Ttr-Cre$^{Tam}$ and *Arid1a*$^{flox/flox}$/Ttr-Cre$^{Tam}$male mice were given a tamoxifen diet (M-Z, low phytoestrogen +1000 mg/kg TAM citrate, SSNIFF, Soest, Germany) for 4 days. These mice are referred to as [*Apc-Arid1a*]$^{ko-TOTAL}$, [*Apc*]$^{ko-TOTAL}$, and [*Arid1a*]$^{ko-TOTAL}$ mice.

Mice were housed under conventional conditions and all reported animal procedures were carried out according to French government regulations (Ethics Committee of Descartes University, Paris). The animal welfare assurance number is APAFIS#14472.

## Immunohistochemistry and in situ hybridization experiments

After sacrifice, livers were harvested, fixed overnight in 4% formalin buffer, and embedded in paraffin. FFPE liver sections were treated as previously described for immunocytochemistry and HE stainings (*de La Coste et al., 1998*). Antibodies used are listed in the Key Resources Table.

RNA in situ hybridization was done on freshly cut 7 µm FFPE liver or kidney sections using the RNAScope 2.5 HD Duplex Kit, with HybEZ II hybridization system, following the manufacturer's instructions (Advanced Cell Diagnostics). The following RNAscope probes were used: Epo (Mm-Epo-C2, Cat. 315501-C2, NM_007942.2, region 39–685), Axin2 (Mm-Axin2, Cat. 400331, NM_015732.4, region 330–1287), DapB (negative control, Cat. 320751, CP015375.1, region 2252107–2252555), Polr2a (positive control, Mm-Polr2a, Cat. 320761, NM_001291068.1, region 3212–4088).

## Hematological analysis and red blood cell counts

Hematological parameters were measured using a CoulterMAXM automatic analyzer (Beckman Coulter) as previously described (*Mastrogiannaki et al., 2009*).

## Plasma collection and ELISA for erythropoietin

At sacrifice, peripheral blood was collected from the inferior vena cava with a heparinized needle (Sigma Aldrich – H3393-50KU). Plasma samples were stored at −80˚C. Plasma EPO protein levels were determined using a Quantikine mouse EPO enzyme-linked immunosorbent assay kit (R and D systems – MEP00B), according to the manufacturer's instructions.

## Treatment with anti-erythropoietin blocking serum

One-year-old [*Apc-Arid1a*]$^{ko-focal}$ and control mice were injected with anti-erythropoietin rabbit serum, as previously described (*Mastrogiannaki et al., 2012*), with minor modifications: injections were performed for 7 consecutive days and mice were sacrificed 18 hr after the last injection. The dose injected was described as able to neutralize a 10-fold excess of circulating erythropoietin (*Mastrogiannaki et al., 2012*). At sacrifice, liver and spleen were collected for immunochemistry and cytometry analysis.

## Hepatocyte isolation and cell culture

Livers from 3-month-old mice were perfused 7 days after the beginning of the tamoxifen diet (1000 mg/kg) with collagenase. The liver cell suspension was collected, and hepatocytes were separated from NPCs by centrifugation for 2 min at 48 g as previously described (*Anson et al., 2012*). The supernatant containing the NPCs was collected and centrifuged for 10 min at 440 g. Hepatocytes were plated as previously described (*Gougelet et al., 2014*; *Torre et al., 2011*; *Guidotti et al., 2003*). Hepa1-6 hepatoma cell line was a gift from C. Perret's lab, authenticated by its *CTNNB1* mutation, assessed by Sanger sequencing. It was tested negative for mycoplasma contamination.

Cells were plated at $3 \times 10^5$ cells per well, in six-well plates, in DMEM solution supplemented with 10% fetal bovine serum, 1% penicillin-streptomycin and fungizone.

Cryopreserved human hepatocytes were obtained from Triangle Research Laboratory (Lonza). They were seeded at confluency ($2.1 \ 10^5$ cell/cm$^2$) and cultured in a humidified 5% $CO_2$ atmosphere at 37°C in hepatocyte growth medium (HGM: WME medium supplemented with 5 µg/ml insulin, 0.1 µM hydrocortisone, 10 µg/ml transferrin, 250 µg/ml ascorbic acid, 3.75 mg/ml fatty-acid-free bovine serum albumin, 2 mM glutamine, penicillin and streptomycin).

## Cell transfection, stimulation, and luciferase assays

Primary murine hepatocytes were transfected with 20 nM small-interfering RNA (siRNA) directed against Arid1a (Qiagen SI00230405) or control siRNA (Dharmacon D-001210-01-05) in the presence of Lipofectamine 2000 (Thermo Fisher Scientific). The next day, cells were stimulated, or not, with 100 ng/ml recombinant mouse Wnt3a (1324-WN) and 100 ng/ml recombinant mouse R-Spondin 3 Protein (4120-RS) (R and D Systems). Molecular analyses were performed 48 hr after transfection or stimulation.

HEPA 1.6 cells were transfected for 24 and 48 hr with 20 nM siRNA directed against *Arid1a* or β-catenin (QiagenSI00942039) or control siRNA. Molecular analyses were performed 72 hr after the first transfection.

Adherent primary human hepatocytes were transfected with 20 nM non-targeting siRNA or siRNAs specific for *APC* (Dharmacon, Lafayette, CO) or *ARID1A* (Qiagen 1027416) at day 1 and day 3 after seeding, using Lipofectamine RNAiMAX (Life Technologies, Carlsbad, CA).

For luciferase assay, primary mouse hepatocytes were transfected using Lipofectamine 2000 (ThermoFisher Scientific) with 1 µg of a luciferase reporter driven by erythropoietin 3' enhancer region (Epo-3'E, 50 nucleotides) (*Huang et al., 1996*), and/or 500 ng of a Renilla vector (Promega, Madison, WI). Luciferase activity was measured 48 hr after transfection with the Dual-Luc kit, according to manufacturer's protocols (Promega).

## Isolation of peliosis-like areas from paraffin-embedded (FFPE) tissue sections and Affymetrix microarrays

Healthy and peliosis-like areas were isolated from 15 to 20 paraffin sections (10 µm) using a small needle under a binocular magnifying glass. After deparaffinization, FFPE tissues were lysed for 24 hr in tissue lysis with proteinase K (Qiagen) at 60°C. Microarray transcriptomic analysis from paraffin-embedded (FFPE) tissue sections was performed on the MTA-31461 chip. Gene set enrichment analysis (GSEA) was performed using the Java tool application available at the Broad Institute (Cambridge, MA, USA). The analysis was performed using Hallmark gene data sets.

## RNA extraction and quantitative RT-PCR

Total RNA was extracted with Trizol reagent (Thermo Fisher Scientific) as previously described (*Gougelet et al., 2016*). Reverse transcription was performed from 100 ng RNA with a cDNA synthesis kit from Thermo Scientific (K1642). The Taqman assays (Thermo Fisher Scientific) and the sequences of PCR primers (Eurogentec) for SybrGreen assays are described in the Key Resources Table. qPCR was performed in duplicate on a LightCycler480 apparatus and the results, analyzed by the ΔΔCt technique, expressed relative to those for 18S RNA.

## Calculation of gene inactivation efficiencies

Arid1a and Apc mRNAs were analyzed by RT-qPCR. For each gene, we used two distinct Taqman assays: (1) One contained two primers both located in undeleted regions. It allowed to detect both wild type and inactivated genes, so the relative mRNA expression of 'TOTAL' gene; (2) In the other, one primer was located in the deleted region. Thus, this Taqman assay allowed to detect and amplify only the 'non excised' gene. We quantified the percentage of inactivation as follows: % of gene inactivation = (1- (mRNA expression of TOTAL *gene expression*/mRNA expression of undeleted *gene expression*)) x 100.

## Protein extracts and western blotting

Livers were lysed mechanically in RIPA buffer (Sigma Aldrich – R0278-50ml) with protease inhibitors (Roche - 11697498001), and boiled in Laemmli sample buffer (Sigma Aldrich – S3401-1VL). 50 µg of protein per lane were run on 8% polyacrylamide gels. The resulting protein bands were electrotransferred onto a 0.2 µm nitrocellulose membrane (Biorad 162–0112), which was then blocked with 5% blocking reagent (Biorad 170–6404) in TBS/Tween 0.1% for 1 hr at RT, probed overnight with the primary antibody, and then incubated with IgG HRP-conjugated secondary antibody for detection with the Clarity ECL substrate (Biorad 70–5061).

To analyze nuclear protein extracts, livers were lysed in Hepes 10 mM pH7.9, KCl 10 mM, EDTA 0.1 mM, EGTA 0.1 mM, DTT 1 mM, AEBSF 0.5 mM. Then, after addition of 12,5 µl of NP40 20% and centrifugation, the pellet was resuspended in Hepes 20 mM pH7.9, NaCl400 mM, EDTA 1 mM, EGTA 1 mM, DTT 1 mM, AEBSF 1 mM, PIC1X, Glycerol 5%, and the supernatants boiled in Laemmli. We next ran 70 µg of protein per lane on Bolt 4–12% Bis-TrisPlus Gels (Thermo Fisher, NW04125BOX). Detection was performed by using Super Signal West Dura ECL system (Thermo Fisher, 34076).

## Electrophoretic mobility shift assay (EMSA)

Nuclear proteins preparation and LXR electromobility shift assay (EMSA) were performed as previously described (*Bobard et al., 2005*). The probes are listed in the Key Resources Table. The HRE element from the Epo-3'E is constituted from two direct repeats of GG/AGTCA sequences with a spacing of two nucleotides (thereafter called DR2).

## Flow cytometry and c-forming unit-erythroid (CFU-E) assays

Primary mouse bone marrow, spleen cells, and NPC liver cells were harvested from [*Apc-Arid1a*]$^{ko-focal}$ mice and their wild-type littermates and erythroid cell populations were identified and analysed using CD71/TER119 flow-cytometric assay. Staining was performed in a 96-well plate and samples ($5.10^4$ cells) were washed once in PBS, 0.4% BSA, 0.1% Sodium Azide, sample staining volume was 50 µl of mix primary-antibody solution, to a final concentration $1.0 \times 10^6$ cells/ml. Primary antibody staining mix were prepared for CD71-FITC and Ter119-PE. Unstained cells, Isotype Ig and single stained cells were used as control and to define boundaries between negative and positive cell labelling. After incubation in the primary antibody stain, two washes were performed by adding 200 µl of staining buffer to each sample.

For CFU-E formation, we plated in duplicate $2 \times 10^5$ bone marrow cells or $2 \times 10^6$ splenic/NPC liver cells in MethoCult M3234 (StemCell Technologies), supplemented, or not, with 2 U EPO. The number of CFU-E colonies was counted after 3 days.

## Chromatin immunoprecipitation (ChIP) and ATAC-qPCR assays

ChIP assay was previously described for hepatocytes isolated after collagenase perfusion in *Gougelet et al., 2014*. Chromatin was immunoprecipitated using 3 µg antibody preabsorbed onto 60 µl protein G agarose (Thermo Fisher Scientific – 10004D). Bindings were assessed on the Axin2 intronic enhancer and hepatic Epo enhancer, relative to that of the immunoglobulin isotype control, by Taqman assay and SYBR green technology, respectively with the following oligonucleotides (Eurogentec): negative control region and hepatic Epo enhancer. Enrichment by ChIP was assessed relative to the input DNA and normalized to the level of the negative controls.

ATAC-qPCR assays were done using omni-ATAC as described in *Corces et al., 2017*, on frozen liver samples after isolation of nuclei. Then, 50,000 nuclei were used for transposition for 30 min in 50 µl reaction mix containing 2.5 µl transposase (Illumina kit #FC-121–103), digitonin and tween 20 at 0.1%. After transposition, the following steps were done according to the initial protocol (*Buenrostro et al., 2015*). The qPCR step was similar to ChIP experiments.

## Statistics

We assessed statistical significances with GraphPad Prism six software. The data represent the mean ± SEM and p values were calculated by two-tailed unpaired Student's t-test, one-way ANOVA, or two-way ANOVA as specified in the figure legends. $p < 0.05$ was considered statistically significant and exact p-values are mentioned unless ****$p < 0.0001$. Each quantitative experiment was repeated

at least three times. We considered biological replicates as those animals or tissues subjected to the same experimental test, and technical replicates as individual samples or tissues subjected to the same analysis.

## Acknowledgements

This work was supported by the French National League against Cancer (LNCC), by the IDEX 'Epiliv-can', by the Institut National du Cancer 'Epigenetics and Liver Cancer', and by the Plan-Cancer Programme « CHROMA-LIV ». RR got fellowships from the French Laboratory of Excellence program 'Who am I ?" (no ANR-11-LABX-0071 included in the Investments for the Future program n° ANR-11-IDEX-0005–01), and the French Foundation for Cancer Research (ARC). We are thankful to Dr C Peyssonnaux team for discussions on erythropoietin expression, and help to measure hematological blood parameters. We are thankful to Dr P Mayeux, JC Deschemin and Dr E Huang for the gift of anti-EPO blocking serum, DFO and EpoE-LUC plasmid, respectively. We wish to thank the animal housing facility at Cochin Institute, and the 'GENOM'IC' facility for transcriptomic data generation and analysis. We are very grateful to Dr C Desbois-Mouthon, Dr S Vaulont and Pr J Weitzman for critical reading of the manuscript, and to Pr J Weitzman, Pr J Zucman-Rossi and Dr C Desdouets for helpful discussions.

## Additional information

### Funding

| Funder | Grant reference number | Author |
| --- | --- | --- |
| Institut National Du Cancer | Epigenetics and Liver Cancer | Rozenn Riou<br>Angélique Gougelet<br>Cécile Godard<br>Julien Calderaro<br>Sabine Colnot |
| Ligue Contre le Cancer | Equipe Labellisée | Rozenn Riou<br>Angélique Gougelet<br>Cécile Godard<br>Sabine Colnot |
| Agence Nationale de la Recherche | Labex "Who Am I" | Rozenn Riou<br>Angélique Gougelet<br>Cécile Godard<br>Sabine Colnot |
| Institut National Du Cancer | Chromaliv | Rozenn Riou<br>Angélique Gougelet<br>Cécile Godard<br>Sabine Colnot |
| Agence Nationale de la Recherche | Idex "EpilivCan" | Rozenn Riou<br>Angélique Gougelet<br>Cécile Godard<br>Sabine Colnot |

The funders had no role in study design, data collection and interpretation, or the decision to submit the work for publication.

### Author contributions

Rozenn Riou, Conceptualization, Formal analysis, Validation, Investigation, Visualization; Meriem Ladli, Validation, Investigation, Visualization; Sabine Gerbal-Chaloin, Formal analysis, Supervision, Validation, Visualization; Pascale Bossard, Formal analysis, Supervision, Validation, Investigation, Visualization; Angélique Gougelet, Julien Calderaro, Formal analysis, Investigation; Cécile Godard, Isabelle Lagoutte, Franck Lager, Investigation; Robin Loesch, Alexandre Dos Santos, Investigation, Involved in the revision process; Zhong Wang, Resources; Frédérique Verdier, Formal analysis, Supervision, Validation, Investigation, Methodology; Sabine Colnot, Conceptualization, Data curation, Formal analysis, Supervision, Funding acquisition, Validation, Investigation, Visualization, Methodology

## Author ORCIDs
Zhong Wang (iD) http://orcid.org/0000-0002-8720-4609
Sabine Colnot (iD) https://orcid.org/0000-0002-3949-9107

## Ethics

Animal experimentation: This study was performed in strict accordance with the French government regulations. All of the animals were handled according to approved institutional animal care and use committee (Ethics Committee of Descartes University, Paris). The protocol was approved by the Ethics Committee of Descartes University, Paris (permit number APAFIS#14472). Every effort was made to minimize suffering.

## Decision letter and Author response
Decision letter https://doi.org/10.7554/eLife.53550.sa1
Author response https://doi.org/10.7554/eLife.53550.sa2

## Additional files

### Supplementary files
- Transparent reporting form

### Data availability

Microarrays have been deposited in GEO database (GSE134553) and are publicly available. All data generated or analysed during this study are included in the manuscript and supporting files. Source data excel files have been provided for Figures 1, 2, 3, 4, 5, 7, 8, 1S1,1S3,3S1,5S1,5S2,7S1.

The following dataset was generated:

| Author(s) | Year | Dataset title | Dataset URL | Database and Identifier |
|---|---|---|---|---|
| Colnot S, Riou R | 2020 | Expression data from isolated areas from [Apc-Arid1a]ko-focal liver tissues after FFPE treatment | https://www.ncbi.nlm.nih.gov/geo/query/acc.cgi?acc=GSE134553 | NCBI Gene Expression Omnibus, GSE134553 |

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
