## [Decision Letter]

**Acceptance summary:**

Riou et al. describe how the *Arid1a* subunit of the SWI/SNF chromatin remodelling complex cooperates with activated Wnt signalling to induce re-expression of the erythropoeitin (EPO) hormone in liver hepatocytes. The overexpressed EPO induces increased red blood cell production in the spleen leading to erythrocytosis and local angiogenesis in the liver. *Arid1a* regulates binding of the TCF4-β catenin complex to an Hnf4a response element in the 3' enhancer of the EPO gene that to promote its expression upon activation of Wnt signalling.

**Decision letter after peer review:**

Thank you for submitting your article "ARID1A loss in adult hepatocytes unleashes β-catenin-mediated erythropoietin transcription" for consideration by *eLife*. Your article has been reviewed by three peer reviewers, including Irwin Davidson as the Reviewing Editor and Reviewer #1, and the evaluation has been overseen by Didier Stainier as the Senior Editor. The following individual involved in review of your submission has agreed to reveal their identity: Tom Bird (Reviewer #3).

The reviewers have discussed the reviews with one another and the Reviewing Editor has drafted this decision to help you prepare a revised submission.

Summary:

This paper describes a potential mechanism by which the SWI/SNF subunit *Arid1a* cooperates with activated Wnt signaling to induce re-expression of the EPO hormone in liver hepatocytes. This study provides new data on how Wnt signaling can regulate EPO expression and on an interplay between *Arid1a*, chromatin accessibility and Wnt signaling. While the reviewers found this study to be of significant interest, a number of major issues should be addressed in a revised version of the paper.

Essential revisions:

1) In Figure 9, the authors summarize their data and proposes a model for how Arid1A regulates TCF4-b-cat activation of the EPO gene. It would be essential to perform ATAC-seq or at least ATAC-qPCR to demonstrate changes in accessibility of the EPO regulatory element under the different conditions. The model proposes that in wild-type conditions and in APC-mutant conditions the SWI/SNF complex is bound at the regulatory element and that it is released under conditions where *Arid1a* is deleted. This model can be easily tested using antibodies that ChIP components of the SWI/SNF complex. The best would be *Arid1a* in combination with other subunits, but if there are no ChIP-grade *Arid1a* antibodies, the authors could at least determine whether SWI/SNF is present under these conditions and released upon *Arid1a* inactivation. One could also imagine other models where SWI/SNF without *Arid1a* is also bound when the element is activated, but with different functional outcome in the presence or absence of Ardi1a. Additional ATAC or ChIP experiments could easily discriminate these possibilities. It is essential that the authors address these issues.

2) The data for physiologically relevant production of Epo by hepatocytes in the model must be more convincingly shown. The absence of systemic/hepatic and renal hypoxia should be more robustly demonstrated. The production of physiologically meaningful Epo by hepatocytes should be shown. It is proposed that the 2-4-fold increase in transcript/reporter observed in cell lines is responsible for the 2-3-fold increase in systemic Epo; most of which is derived from kidney physiologically. Comparing kidney as positive control at transcript level would be helpful as well as measurement at protein level in liver and kidney. The authors should quantify and compare hepatic and kidney EPO expression in the different genetic backgrounds. This could be done by both qRT-PCR and by RNA-fish/RNAscope on sections from both kidney and liver. It is essential the authors quantify relative EPO expression in both organs.

3) The authors should address if the observed peliosis is related to sinusoidal obstruction related to the polycythaemia. If this is the case then this would be consistent with the splenomegaly – due to portal hypertension. Is there flow in the peliotic regions on ultrasound? An alternative explanation for the results in this paper would be that a combination of Arid1/APC deletion results in sinusoidal constriction/obstruction with pooling/extravasation of blood. This constriction would then lead to portal hypertension and increased Epo release by the kidney resulting in polycythaemia and peliosis. This seems a potential explanation and is not refuted by the data presented.

4) Peliosis and HCC have been described together in HCC but this is not referenced e.g. doi: 10.1007/s00534-008-0035-9. Furthermore, interrogation of publicly available datasets (e.g. TCGA) may add strength to the proposition that Wnt pathway mutations in combination with Arid1A give rise to Epo production within the liver tumour.

5) A statement regarding the cause of death in APC/Arid1 animals should be provided to support the authors claim that this is secondary to peliosis. This is not normally itself a fatal disorder, so why do the authors propose it is in this model? Similarly, is there correlation between the ultrasound findings and endpoint liver examination at post mortem?

6) In line with general recommendations in mouse work it would be important to show that the key findings are present in both male and female model systems.

7) Additional replicates are necessary to show absence of effects of HIF targets. There do appear to be differences in at least some of these and it is likely that n=4 will be insufficient to demonstrate equivalence.

[Editors' note: further revisions were suggested prior to acceptance, as described below.]

Thank you for resubmitting your work entitled "ARID1A loss in adult hepatocytes activates β-catenin-mediated erythropoietin transcription" for further consideration by *eLife*. Your revised article has been evaluated by Didier Stainier (Senior Editor) and a Reviewing Editor.

The manuscript has been improved but there are some remaining issues that need to be addressed before acceptance, as outlined below:

The manuscript of Riou et al. has been substantially modified and reorganised. The authors have performed the majority of the essential revisions, in particular the RNAscope analyses of EPO expression in the wild-type and mutant livers and kidneys. The only experiment that could not be performed was the analyses of the phenotype of the female mice that is likely to take a much longer time due to the COVID crisis that restricted access and breeding in the animal facility.

While the manuscript is very much improved, several issues still require attention and should be addressed before final acceptance.

1) In the rebuttal letter and in the revised text, the authors describe the EPO expression seen by RNAscope experiments as “bursty” or “stochastic”. While this may be the case, it is not fully justified, nor necessary to explain the phenotype to describe EPO expression in this way. The text should be modified.

2) The ATAC-qPCR experiments did not reveal increased chromatin accessibility upon *Arid1a* inactivation, in fact, chromatin accessibility appeared to decrease compared to Apc inactivation alone. It appears that only a single primer set was used that overlaps with the TCF binding site. From this limited analysis, it is difficult to assess exact nucleosome positioning and how it may be affected by *Arid1a* loss. It would have been more appropriate to test several amplicons spanning the enhancer. This would have perhaps better revealed how *Arid1a* loss affects nucleosome positioning and accessibility.

3) Last but not least, the quality of the English in the revised version is very poor leading to lack of precision and making the text difficult to follow. A thorough revision of the text by a native speaker is essential.

---

## [Author Response]

Essential revisions:1) In Figure 9, the authors summarize their data and proposes a model for how Arid1A regulates TCF4-b-cat activation of the EPO gene. It would be essential to perform ATAC-seq or at least ATAC-qPCR to demonstrate changes in accessibility of the EPO regulatory element under the different conditions. The model proposes that in wild-type conditions and in APC-mutant conditions the SWI/SNF complex is bound at the regulatory element and that it is released under conditions where Arid1a is deleted. This model can be easily tested using antibodies that ChIP components of the SWI/SNF complex. The best would be Arid1a in combination with other subunits, but if there are no ChIP-grade Arid1a antibodies, the authors could at least determine whether SWI/SNF is present under these conditions and released upon Arid1a inactivation. One could also imagine other models where SWI/SNF without Arid1a is also bound when the element is activated, but with different functional outcome in the presence or absence of Ardi1a. Additional ATAC or ChIP experiments could easily discriminate these possibilities. It is essential that the authors address these issues.

We agreed with this essential requirement. We had tried ChIP-qPCR experiments using antibodies against Arid1a or against Brg1. But primary mouse hepatocytes are difficult to study by ChIP and this previously took us time to implement ChIP against Tcf4 and/or β-catenin (see Gougelet et al., 2014). Consequently, we got no convincing data using these antibodies. We thus undertook ATAC-qPCR experiments using OMNI-ATAC recently described technique, which works on frozen liver tissue. We clearly found that chromatin accessibility increased after β-catenin activation on GS, Axin2 and Epo enhancer. But surprisingly, this was not the case when Arid1a gene is inactivated. This goes well with the fact that Arid1a is a facultative component of BAF SWI/SNF chromatin complex, which can be replaced by Arid1b. This has been added in Figure 8F, leading to a modified model of transcription of Epo shown in Figure 9. This is described in the text: “Tcf4 bound in vivo to the *Epo* enhancer and such binding was slightly higher in [*Apc*]^ko-TOTAL^ and much higher in [*Apc-Arid1a*]^ko-TOTAL^ hepatocytes than in controls (Figure 8E). After a single β-catenin activation process, the H3K27me3 repressive mark slightly decreased and chromatin was more accessible (Figure 8F). In contrast, the loss of Arid1a strongly decreased H3K27Me3 repressive mark without modifying chromatin access. In [*Apc-Arid1a*]^ko-TOTAL^ hepatocytes, a H3K27Ac active histone mark was induced while chromatin accessibility was lower.” This is also discussed: “For Epo enhancer, the increased binding of Tcf4/β-catenin complex to the Hnf4-RE is dependent on *Arid1a*, whose loss decreases H3K27me3 repressive mark. That could be due to the complex existing balance between the Polycomb complex PRC2 (catalyzing the addition of methyl groups to histone H3 at lysine 27) and the SWI/SNF complex (Kadoch, Copeland and Keilhack, 2016). Accordingly, the inhibition of the Polycomb EZH2 subunit is synthetically lethal in ARID1A-mutated tumors (Bitler et al., 2015; Allredge and Eskander, 2017). Therefore, Arid1a and Polycomb complex could act in concert to modulate liver expression of the *Epo* gene”.

2) The data for physiologically relevant production of Epo by hepatocytes in the model must be more convincingly shown. The absence of systemic/hepatic and renal hypoxia should be more robustly demonstrated. The production of physiologically meaningful Epo by hepatocytes should be shown. It is proposed that the 2-4-fold increase in transcript/reporter observed in cell lines is responsible for the 2-3-fold increase in systemic Epo; most of which is derived from kidney physiologically. Comparing kidney as positive control at transcript level would be helpful as well as measurement at protein level in liver and kidney. The authors should quantify and compare hepatic and kidney EPO expression in the different genetic backgrounds. This could be done by both qRT-PCR and by RNA-fish/RNAscope on sections from both kidney and liver. It is essential the authors quantify relative EPO expression in both organs.

Here again, the reviewers indicated some useful experiments to perform. We first assessed Epo mRNA expression by qPCR in the transgenic kidneys, and showed that there is no increase in renal Epo expression after either focal (new Figure 2E) or panlobular (new Figure 5D) Apc/Arid inactivations. In this latter case, there is rather a slight decrease of renal Epo after hepatic Apc/Arid knock-out. This clearly showed that the systemic impact of Epo is due to hepatic de novo transcription. We also performed. In this context, measuring renal hypoxia was not of interest. We focused on hepatic localization of Epo transcripts. After some attempts to perform classical in situ hybridizations, even if we are expert in mRNA ISH in the liver (Benhamouche et al., 2006), they failed when assessing Epo transcripts. We thus performed highly sensitive RNAScope experiments, and as expected we showed: (1) that Epo mRNA are only detectable in rare renal interstitial cells of normal kidney; (2) that no Epo is detected in normal livers (Figure 6—figure supplement 1). In transgenic livers, we were able to detect Epo transcripts only in rare Axin2-expressing (and thus β-catenin-activated) hepatocytes of Apc/Arid knock-out livers. This exciting experiment clearly underlines a stochastic and bursty transcription of Epo as described previously for several hepatic mRNAs by Itzkovitz’ team using single-RNA ISH (Halpern, Nature 2015).

3) The authors should address if the observed peliosis is related to sinusoidal obstruction related to the polycythaemia. If this is the case then this would be consistent with the splenomegaly – due to portal hypertension. Is there flow in the peliotic regions on ultrasound? An alternative explanation for the results in this paper would be that a combination of Arid1/APC deletion results in sinusoidal constriction/obstruction with pooling/extravasation of blood. This constriction would then lead to portal hypertension and increased Epo release by the kidney resulting in polycythaemia and peliosis. This seems a potential explanation and is not refuted by the data presented.

In keeping with this hypothesis, we performed dynamic ultrasound using microbubble administration and an example is shown in Figure 1—figure supplement 2. It clearly showed that there is a decrease in hepatic vascular perfusion within echogenic areas enriched in Apc/Arid inactivated hepatocytes. However, no modification in kidney Epo expression was seen, showing that the primary defect resulting in polycythaemia and peliosis is a hepatic one.

4) Peliosis and HCC have been described together in HCC but this is not referenced e.g. doi: 10.1007/s00534-008-0035-9. Furthermore, interrogation of publicly available datasets (e.g. TCGA) may add strength to the proposition that Wnt pathway mutations in combination with Arid1A give rise to Epo production within the liver tumour.

Thank you for the reference about peliosis and HCC, added in the Discussion paragraph eight. We have interrogated TCGA dataset as legitimately proposed. We could not find Epo overexpression in CTNNB1/ARID1A mutated HCCs. However, the frequent hypoxia found in cancers rendered this observation not so surprising. This has been added as a supplemental figure (Figure 1—figure supplement 4) and in the text (subsection “Emergence of peliosis-like regions in the liver of [Apc-Arid1a]^ko-focal^ mice”) and commented in the Discussion.

5) A statement regarding the cause of death in APC/Arid1 animals should be provided to support the authors claim that this is secondary to peliosis. This is not normally itself a fatal disorder, so why do the authors propose it is in this model? Similarly, is there correlation between the ultrasound findings and endpoint liver examination at post mortem?

To clarify this issue, we have added in Figure 1D a picture of a necropsy from a Apc/Arid 13-month old mouse whose entire liver was diseased with a dramatic erythrocytosis and no healthy area in its liver. This is representative of the 3 mice necropsied before dying, as they were reaching the endpoints: “At necropsy of dying mice, the whole liver was diseased, appearing dark red, filled with blood, with large necrotic areas, and no remaining healthy zones (Figure 1D, inset)”.

6) In line with general recommendations in mouse work it would be important to show that the key findings are present in both male and female model systems.

We agree with this comment. We did all the analyses in males because the initial question was to analyze liver carcinogenesis for which an important delay in females is described. We tried to generate new female mice with the different genotypes. We got no enough female mice with the 4 genotypes expected in March. Unfortunately, we could not continue the breedings due to the COVID19 crisis as the access to the animal facility has been severely restricted.

7) Additional replicates are necessary to show absence of effects of HIF targets. There do appear to be differences in at least some of these and it is likely that n=4 will be insufficient to demonstrate equivalence.

We agree with this comment and have expanded this analysis to n=8 (n=7 for Arid ko livers). There are some differences in the expression of Eno2, Car9 and Rab42 mRNA expressions, which were seen in Apcko and Apc/Aridko livers (Figure 7—figure supplement 1D-E). This underlines an overexpression of some HIF targets after β-catenin activation, and does not emphasize that Apc/Arid-ko livers are specifically HIF-activated. This is written in the text: “A small subset of Hif1α/Hif2α targets was slightly overexpressed in both Apc^ko^ and Apc-Arid1a^ko^ livers, such as Eno2, Car9 and Rab42, confirming that β-catenin and HIF signaling share some transcriptional targets (Figure 7—figure supplement 1D,E) ( Benhamouche et al., 2006).”

[Editors' note: further revisions were suggested prior to acceptance, as described below.]

The manuscript of Riou et al. has been substantially modified and reorganised. The authors have performed the majority of the essential revisions, in particular the RNAscope analyses of EPO expression in the wild-type and mutant livers and kidneys. The only experiment that could not be performed was the analyses of the phenotype of the female mice that is likely to take a much longer time due to the COVID crisis that restricted access and breeding in the animal facility.While the manuscript is very much improved, several issues still require attention and should be addressed before final acceptance.1) In the rebuttal letter and in the revised text, the authors describe the EPO expression seen by RNAscope experiments as “bursty” or “stochastic”. While this may be the case, it is not fully justified, nor necessary to explain the phenotype to describe EPO expression in this way. The text should be modified.

We agree that our interpretation of RNAscope experiments was hypothetic. We have suppressed these words, and have only discussed the interest of single-RNA *in situ* hybridization approach to better describe gene expression in the liver, as it was done by Itzkovitz’ lab.

2) The ATAC-qPCR experiments did not reveal increased chromatin accessibility upon Arid1a inactivation, in fact, chromatin accessibility appeared to decrease compared to Apc inactivation alone. It appears that only a single primer set was used that overlaps with the TCF binding site. From this limited analysis, it is difficult to assess exact nucleosome positioning and how it may be affected by Arid1a loss. It would have been more appropriate to test several amplicons spanning the enhancer. This would have perhaps better revealed how Arid1a loss affects nucleosome positioning and accessibility.

We limited the chromatin accessibility analysis due to the difficulty to design accurate PCR primers in the region of the enhancer. For this revision, we finally designed three additional primer pairs which work for qPCR (added in the key resources table), but we could not technically try amplicons in the flanking regions, nor get an amplicon specific for the Hnf4/Wnt responsive element (see Figure 8—figure supplement 2A). We revealed an increased chromatin accessibility when the amplicon is centered on the Hnf4/Wnt responsive element. These data are added as Figure 8—figure supplement 2 and commented in the legend.

3) Last but not least, the quality of the English in the revised version is very poor leading to lack of precision and making the text difficult to follow. A thorough revision of the text by a native speaker is essential.

The text of the Results and the Discussion has been thoroughly revised by a professional scientific proofreader and editor.